# Heterogeneous ice nucleation on dust particles sourced from nine deserts worldwide - Part 2: Deposition nucleation and condensation freezing

Yvonne Boose[1,2], Philipp Baloh[3], Michael Plötze[4], Johannes Ofner[5], Hinrich Grothe[3], Berko Sierau[1], Ulrike Lohmann[1], and Zamin A. Kanji[1]

[1]Institute for Atmospheric and Climate Science, ETH Zürich, Zürich, Switzerland
[2]now at: Institute of Atmospheric Physics, German Aerospace Center, Wessling, Germany
[3]Institute for Materials Chemistry, TU Wien, Vienna, Austria
[4]Institute for Geotechnical Engineering, ETH Zürich, Zürich, Switzerland
[5]Institute for Chemical Technologies and Analytics, TU Wien, Vienna, Austria

**Correspondence:** Y. Boose (yvonne.boose@alumni.ethz.ch) and Z. A. Kanji (zamin.kanji@env.ethz.ch)

**Abstract.** Mineral dust particles from deserts are amongst the most common ice nucleating particles in the atmosphere. The mineralogy of desert dust differs depending on the source region and can further fractionate during the dust emission processes. Mineralogy to a large extent explains the ice nucleation behavior of desert aerosol, but not entirely. Apart from pure mineral dust, desert aerosol particles are often mixed with small amounts of biological material or particles exhibit a coating. Aging on the ground or during atmospheric transport can deactivate nucleation sites and thus strong ice-nucleating minerals may not exhibit their full potential. In the partner paper of this work, it was shown that mineralogy determines most but not all of the ice nucleation behavior in the immersion mode found for desert dust. In this study, the influence of semi-volatile organic compounds and the presence of crystal water on the ice nucleation behavior of desert aerosol is investigated. This work focuses on the deposition and condensation ice nucleation modes at temperatures between 238 and 242 K of 18 dust samples sources from 9 deserts worldwide. Chemical imaging of the particles' surface is used to determine the cause of the observed differences in ice nucleation. It is found that while the ice nucleation ability of the majority of the dust samples is dominated by their quartz and feldspar content, in one carbonaceous sample it is mostly caused by organic matter, potentially cellulose and/or proteins. On the other hand, the ice nucleation ability of an airborne Saharan sample is found to be diminished, likely by semi-volatile species covering ice nucleation active sites of the minerals. This study shows that in addition to mineralogy, other factors such as organics and crystal water content can alter the ice nucleation behavior of desert aerosol during atmospheric transport in various ways.

## 1 Introduction

The ice phase in clouds causes one of the largest uncertainties for understanding the role of clouds in the present climate and for projecting future climate (Boucher et al., 2013). While it is known that for the initial formation of ice in clouds warmer than 235 K certain aerosol particles, so-called ice nucleating particles (INPs), are necessary, many aspects of heterogeneous ice

nucleation remain poorly understood (Coluzza et al., 2017; Kanji et al., 2017). Mineral dust is thought to be the most prevalent INP type in the atmosphere (Hande et al., 2015). Schaefer (1949) found naturally occurring mineral dust particles to nucleate ice at temperatures $T < 258$ K. Cozic et al. (2008) and Kamphus et al. (2010) detected mineral dust in ice crystal residuals in mixed-phase clouds. In these clouds the most common ice-nucleating mechanisms are likely immersion and contact freezing. Both mechanisms require a cloud droplet to form first and an INP to either initiate freezing from the inside of the droplet (immersion) or via contact with the surface of the droplet (contact freezing). Cziczo et al. (2013) observed mineral dust also in ice crystal residuals in cirrus clouds. When the relative humidity with respect to ice is high enough (e.g. $RH_i > 140$ % at 238 K), ice forms via homogeneous freezing of solution droplets without the help of an INP (Koop et al., 2000). At lower $RH$ ice may form via immersion freezing on INPs in solution droplets (Zuberi et al., 2002) or via the deposition mode, where ice nucleation occurs on an INP directly from the vapor phase (Vali, 1985; Vali et al., 2015). Condensation freezing is understood as freezing occurring during the formation of a liquid phase, when water saturation is exceeded but before a droplet has formed. Recently, the differentiation between condensation and immersion freezing has been questioned (Marcolli, 2014; Vali et al., 2015). Furthermore, it has been suggested that freezing at water subsaturated ($RH_w < 100\%$) conditions, referred to as deposition nucleation, may in some cases be explained by condensation and subsequent freezing of water in pores on the particles' surface (Pore Condensation and Freezing, PCF, Marcolli 2017).

Mineral dust is thought to have an influence on cloud microphysical processes on a global scale with global dust emission rate estimates of up to 5 Pg yr$^{-1}$ (Engelstaedter et al. 2006 and references therein). DeMott et al. (2003) found increased concentrations of INPs in air masses over Florida which carried Saharan dust, while Creamean et al. (2013) observed precipitation in California to be influenced by dust from Asia and the Sahara. Over Europe, Chou et al. (2011) and Boose et al. (2016a) found periods of Saharan dust advection to coincide with increased INP concentrations under mixed-phase cloud conditions. Even at the South Pole, Kumai (1976) identified about 60% of ice crystal residuals to be clay minerals.

For several decades, clay minerals were believed to be responsible for the ice activity of mineral dust, mainly due to their high mass fraction in airborne dust. Recently, K-feldspars have been identified to nucleate ice at warmer temperatures or lower relative humidity than all other minerals, both in the immersion mode (Atkinson et al., 2013; Zolles et al., 2015) as well as in the deposition and condensation modes (Yakobi-Hancock et al., 2013). Paramonov et al. (2018) found for three dust samples from Iceland, China and the Himalayas the K-feldspar content to correlate well with the ice nucleation activity of dust at temperatures between 238 and 243 K. While Kaufmann et al. (2016) found K-feldspar only in one out of eight dust samples collected in potential atmospheric dust source regions in South America, Asia and Africa, we observed K-feldspar to be present in all but one sample from a collection of 21 samples from deserts around the world (Boose et al., 2016c). Furthermore, feldspars are prone to chemical weathering processes in acids or water which may passivate certain active sites and decrease the feldspar's ice nucleation activity (Augustin-Bauditz et al., 2014; Wex et al., 2014; Harrison et al., 2016). The nature of these active sites is still under debate. It is suspected that they are associated with high energy defects in the lattice structure such as steps, cracks and impurities (Fletcher, 1969; Marcolli et al., 2007) or crystal boundaries in twinned crystals (Harrison et al., 2016) where a (100) crystal plane is exposed to water or vapor (Kiselev et al., 2017). Whale et al. (2017) found that feldspars with perthitic microtexture, i.e. intergrowth of sodic alkali feldspar into a host of K-feldspar, have the highest ice nucleating ability. Berner

and Holdren (1977) observed weathering to occur primarily at excess energy sites on the feldspar surface. During atmospheric transport, such chemical weathering processes or aging could alter the ice nucleation activity of feldspar particles compared to those on the ground or freshly cleaved crystals used in laboratory studies (Atkinson et al., 2013; Yakobi-Hancock et al., 2013; Harrison et al., 2016; Kiselev et al., 2017). By implementing parameterizations for marine organics and feldspar INPs into their global model, Vergara-Temprado et al. (2017) found indications that terrestrial INP concentrations could be dominated

by feldspar. However, atmospheric aging effects were not taken into account. Thus, feldspar has the potential to be the most important ice-nucleating mineral in the atmosphere but its atmospheric relevance is yet to be confirmed.

Atmospheric aging processes are challenging to observe in-situ, thus several laboratory studies have mimicked potential aging processes. These processes often modify the surface of dust particles and as such the ice nucleation ability of mineral dust. Zolles et al. (2015) could block and unblock surface ice nucleation sites with selected organic molecules. Sulfuric acid coating

leads to a reduction in the ice nucleation ability of mineral dust (Sullivan et al., 2010; Augustin-Bauditz et al., 2014), the exposure to low amounts of ozone increases it (Kanji et al., 2013) and coatings of organic aerosol make no difference to it in condensation mode (Koehler et al., 2010; Kanji et al., 2018). The presence of ammonium sulfate has been suggested to improve the ice nucleation ability of Saharan dust advected to Tenerife (Boose et al., 2016b). Recently, Kumar et al. (2018a) and Whale et al. (2018) confirmed that very dilute ammonium salt solutions ($< 1 \ \mathrm{mol \ kg^{-1}}$) increase the ice nucleation temperature of

microcline by 3 to 4.5 K. While surface-collected dust particles from the Sahara were found to have negligible amounts of nitrate or sulfate, a high degree of mixing of nitrate and/or sulfate with mineral dust was observed after advection to Cape Verde, Tenerife or Ireland (Kandler et al., 2007; Dall'Osto et al., 2010).

Residues from ice-nucleating biological material such as fungal proteins or nanoscale pollenaceous INPs have been observed to adsorb to mineral dust while retaining their ice nucleation ability (Schnell, 1977; Conen et al., 2011; Augustin-Bauditz

et al., 2016; O'Sullivan et al., 2015, 2016). Even though desert soils contain typically $< 1 \ \%$ organic matter due to the low average annual precipitation (Troeh and Thompson, 2005), long-range transport of dust has been suggested to efficiently disperse bacteria on a global scale (Hara and Zhang, 2012). Enriched fluorescent particle concentrations, an indication for enriched biological material, in long-range transported Saharan dust were found by Kupiszewski et al. (2015) in ice crystal residuals from mixed-phase clouds in the Swiss Alps, and in condensation mode INPs at 241 K by Boose et al. (2016b) on the Canary

Islands. In contrast to biological material, secondary organic aerosol coatings have been observed to decrease the ice nucleation ability of dust particles in the laboratory in deposition mode (Möhler et al., 2008) but not in condensation mode (Koehler et al., 2010; Kanji et al., 2018).

In a partner paper to this work (Boose et al., 2016c) we investigated the immersion mode ice nucleation activity of airborne dust samples, which were collected after atmospheric transport or sampled from the surface in deserts. We showed that the

K-feldspar fraction, i.e. the fraction of microcline plus orthoclase, of these dust samples correlates well with the ice-active surface site density in the immersion mode at $T = 253$ K. At $T \leq 245$ K the best correlation of the ice nucleation activity was found for the bulk quartz plus feldspar (microcline, orthoclase and plagioclase) content in the dust samples while the fraction of clays was negatively correlated with the ice nucleation activity. Quartz alone has been found to show various immersion mode ice nucleation activities in laboratory studies. Zolles et al. (2015) found quartz being active at temperatures comparable

to microcline, while Atkinson et al. (2013) measured ice nucleation activity below feldspar temperatures but above those of clay. Kaufmann et al. (2016), on the other hand, only observed ice nucleation activity at temperatures comparable to or lower than those of clays. These differences in ice nucleation ability can be related to the history of the quartz samples and different ways of pre-processing them (Zolles et al., 2015). Milling quartz samples leads to a break-up of Si-O-Si bridges on the surface, leading to the formation of Si-OH and Si-O-OH in the presence of water vapor, which increases the ice nucleation activity of

the quartz particles (Kumar et al., 2018b). Quartz is the most abundant mineral on Earth and is widely spread in various soils. It is highly resistant to chemical and mechanical weathering (Goldich, 1938), the latter leading to its abundance increasing with particle size. But quartz is found to a lower degree also in smaller sized dust particles (e.g. 11% volume fraction of 1.6 $\mu$m sized particles over Morocco, Kandler et al. 2009).

The current paper focuses on the ice nucleation behavior at 238 to 242 K of airborne and surface collected dust samples.

We investigate ice nucleation at a constant temperature while $RH$ is increased from ice saturation to above water saturation. While the partner paper (Boose et al., 2016c) showed that mineralogy explains most but not all of the observed ice nucleation behavior of desert dust, this paper focuses on the role of compounds other than the pure minerals for ice nucleation. We use thermogravimetric analysis and chemical imaging methods to highlight the effect of other entities mixed with the dust, such as organic material or soot, on the ice nucleation behavior. In addition to the samples studied in the partner paper (Boose et al.,

2016c), three more airborne Saharan samples are investigated. Comparing the in total seven airborne Saharan samples to in-situ measurements in the Saharan Air Layer, we find low variability in the ice nucleation behavior of dust from different sources. Furthermore, we show that airborne samples containing orthoclase are similarly active at the studied temperatures as those containing microcline.

## 2   Methods

### 2.1   Dust sample origins and processing

In this part of the series we present ice nucleation measurements of 18 dust samples. 7 airborne samples were collected after advection from the Sahara. 4 of the airborne samples were collected directly from the air in August 2013 and 2014 at the Izaña observatory on Tenerife, Spain, using a custom-made large cyclone (Advanced Cyclone Systems, S.A.: flow rate: 200 $\mathrm{m^3h^{-1}}$,

$D_{50}$ = 1.3 $\mu$m, the diameter at which the collection efficiency is 50%). The remaining three airborne samples were collected by deposition on solar panels or roofs in April 2014 (Crete and Peloponnese, Greece) and on 10[th] of May, 2010 (Aburdees, Egypt). Nine samples were collected from the surface in the following deserts: (i) the Atacama desert in Chile; (ii) a location approximately 70 km from Uluru in Australia; (iii) the Great Basin in Nevada and (iv) the Mojave desert in California, USA; (v) a Wadi in the Negev desert, approximately 5 km from Sde Boker in Israel; (vi) dunes in the Sahara, close to Merzouga in Morocco; (vii) dunes in the Arabian desert in Dubai; (viii) the Etosha pan in Namibia, a dry salt pan; and (ix) the Taklamakan desert in China. A map showing the locations is provided in Boose et al. (2016c). Before arriving to the laboratory, samples were stored in various ways: Samples collected from the surface were typically stored for several weeks in PET bottles or other

plastic containers. Airborne samples were stored in polypropylene tubes and sealed with paraffin wax tape. In the laboratory, all samples were stored in the dark at room temperature in polypropylene tubes after pre-processing (sieving/milling, see below). While changes in the ice nucleating ability due to water uptake, loss of volatile material or growth of biological material which may occur during storage cannot be excluded, they are assumed to be minor because the samples were collected and stored under dry conditions, hardly exposed to air and kept at a lower temperature than at which they were collected. The Israel

sample and the Etosha sample are from the same batch as those studied in Kaufmann et al. (2016). The surface-collected samples were sieved with a cascade of dry sieves (Retsch Vibratory Sieve Shaker AS 200) with 32 $\mu$m diameter being the smallest cut-off size. Most samples only contained a few weight percent in this size range. The Australia and Morocco samples were milled using a vibratory disc mill (Retsch, model RS1) as the fraction of particles in the sub-32 $\mu$m size range was too low for ice nucleation experiments. Particles in the lowest available size bin (32 to 64 $\mu$m) of the Morocco sample were milled.

The Australia sample was first sieved with a coarse, millimeter range sieve to separate any large material, and the remaining smaller fraction was milled. For the Israel and the Atacama dusts a sieved and a milled sample was included in the study. The Israel sample was first sieved and part of sub-32 $\mu$m fraction was milled while in case of the Atacama sample part of the initial, unsieved batch was milled.

To investigate if the ice nucleation activity of the samples is influenced by biological particles internally or externally mixed

with the dust or by organic coatings on the dust particle surface, selected samples were heated to 300°C and stayed at this temperature for 10 h before the ice nucleation experiments. At this temperature proteinaceous material, such as bacterial and fungal INPs should be denatured (Pouleur et al., 1992) and the majority of organic material, such as glucose, is combusted and evaporated (Kristensen 1990 and references therein).

## 2.2 Dust particle generation and size distribution

Dust particles were dry dispersed using a rotating brush generator (RBG, Palas, model RBG 1000) with $N_2$ (5.0) as carrier gas into a 2.78 m$^3$ stainless steel aerosol reservoir tank (Kanji et al., 2013) via a cyclone that confined the dust size distribution to below $D_{50}$ = 2.5 $\mu$m. Total particle concentration was monitored using a condensation particle counter (CPC; TSI model 3772). The tank was filled with particles up to a concentration of 1200 cm$^{-3}$, which decreased steadily to about 300 cm$^{-3}$ over approximately 10 h. The tank was cleaned before an experiment by repeatedly evacuating and purging it with $N_2$ until the

particle concentration decreased to 30 - 90 cm$^{-3}$.

The particle size distribution of all samples was measured with a scanning mobility particle sizer (SMPS; TSI; DMA model 3081, CPC model 3010) for mobility diameters ($d_m$) between 12 - 615 nm and an aerodynamic particle sizer (APS; TSI; model 3321) for aerodynamic diameters ($d_{aer}$) between 0.5 - 20 $\mu$m. The mobility and aerodynamic diameters were converted to volume equivalent diameter ($d_{ve}$) by assuming a typical dust particle density of $\rho$ = 2.65 g cm$^{-3}$ (Hinds, 1999; Kandler et al., 2007; Hiranuma et al., 2015a) and optimizing the shape factor $\chi$ to receive the best overlap of two size distributions measured by the SMPS and APS. This yielded $\chi$ = 1.36, which is in the range of earlier studies (Hinds, 1999; Alexander, 2015; Hiranuma et al., 2015a). Assuming spherical particles, the area size distribution was calculated and fitted with a bimodal log-normal distribution. The mean particle surface area ($\overline{A_{ve,w}}$) was calculated from the resulting fit for each sample. Four size distributions

and the fit parameters for all samples are provided in Boose et al. (2016c). During an experiment $\overline{A_{\text{ve,w}}}$ was reduced by between 6 to 24% due to a faster sedimentation of larger particles in the aerosol tank. The Great Basin sample was coarser than the other samples and settled out faster. Therefore two refills were necessary and $\overline{A_{\text{ve,w}}}$ varied by 64%.

## 2.3    Mineralogical, thermogravimetric and morphology analysis

The quantitative mineralogical composition of the bulk dust samples was investigated with the X-ray diffraction (XRD) Rietveld method (Rietveld, 1969) using a Bragg-Brentano diffractometer (Bruker AXS D8 Advance with CoKalpha-radiation). The qualitative phase composition was determined with the software DIFFRACplus (Bruker AXS). On the basis of the peak positions and their relative intensities, the mineral phases were identified in comparison to the PDF-2 data base (International Centre for Diffraction Data). The quantitative composition was calculated by means of Rietveld analysis of the XRD pattern

(Rietveld program AutoQuan, GE SEIFERT, Bergmann et al. 1998; Bish and Plötze 2011). Due to the small amount of dust sample, it was not possible to do a mineralogical analysis of the identical size fraction as in the tank ($< 2.5$ $\mu$m). Instead, the entire size fraction of the airborne and the milled samples, and the sub-32 $\mu$m fraction of the sieved samples was used. The measured mineralogical composition is provided in Boose et al. (2016c) and that of the additional Tenerife samples in Table 1. The Tenerife2014_1 sample was additionally measured by Powder XRD (Panalytical XPert Pro) in Bragg-Brentano geometry

equipped with a Copper anode providing CuK$a$ radiation. A diffractogram was recorded before and after heating the sample to 300°C for 10 h on a silicon sample carrier.

Thermogravimetric analysis (TGA) of six of the dust samples was conducted by gradually heating the dust samples from 40 to 300°C at 10 K/min and continuously recording the mass of the samples in a thermogravimetric analyzer (Model Pyris 1 TGA, PerkinElmer). During the temperature scan, the samples were under a constant nitrogen flow of 20 ml/min.

The morphology of one samples was investigated using scanning electron microscopy (SEM, FEI Quanta 250 FEG, ThermoFisher Scientific).

## 2.4    Chemical imaging: ATR spectrometry and Raman mapping

Infrared-Attenuated Total Reflection (IR-ATR) spectroscopy was carried out on a FTIR (Bruker Vertex 80v) equipped with an

ATR cell (Pike GladiATR, diamond ATR crystal). The beam path of the spectrometer and the optical parts of the ATR cell are under vacuum (1.65 mbar) to minimize the influence of water vapor and $CO_2$. The crystal where the sample is placed sits in a cell that is flushed with nitrogen gas before and through the measurement for the same reason. A liquid nitrogen cooled MCT detector is used for spectra acquisition. Spectra of the samples were recorded before and after a heat treatment. For the heat treatment the samples were placed for 10 h into a laboratory oven at 300°C. The spectral window was set between 700 and 4000 cm$^{-1}$ with a resolution of 1 cm$^{-1}$. Spectra for the Tenerife2014_1, Etosha and Australia samples were recorded this way. Raman images were recorded on a confocal Raman spectrometer (WiTec alpha300 RSA+) using a 488 nm laser, 50x magnification, 600 lines mm$^{-1}$ grating and a laser power of 4.7 mW. For Raman imaging the dust was impacted on a

clean aluminium surface as described in Ofner et al. (2017) and subsequently mapped with the aid of a Piezo XY-stage. Raman mappings were carried out before and after the same heat treatment as for the IR-ATR measurements. The Etosha and Australia samples were mapped by this method.

## 2.5   Deposition and condensation nucleation experiments and data treatment

Ice nucleation experiments were conducted with the portable ice nucleation chamber, PINC (Chou et al., 2011; Boose et al.,
2016a). Aerosol particles are sampled from the tank, dried and introduced into the chamber, where they are layered between two particle-free sheath air flows. Before an experiment, the two chamber walls are coated with a thin layer of ice. During an experiment a temperature gradient is applied between the walls, leading to diffusion of water vapor and heat. A linear gradient of temperature and the partial pressure of water vapor between the walls leads to supersaturation with respect to ice. At a constant aerosol layer temperature, the relative humidity ($RH$) is raised at a constant rate until supersaturation with respect to
water of a few percent ($RH_{\mathrm{w}}$ = 103-105%) is reached. If ice nucleation occurs under these conditions, ice crystals grow on the INPs and are detected in the lower part of the chamber by an optical particle counter. As the ice nucleation mechanisms cannot be identified visually in PINC we refer to the deposition mode at $RH_{\mathrm{w}}$ < 100% and to condensation freezing at $RH_{\mathrm{w}}$ > 100%. We use these relative humidity-based thermodynamic regimes as an operational definition, which does not exclude the possibility of PCF to occur at $RH_{\mathrm{w}}$ < 100% (Marcolli, 2017). Condensation mode refers here to the conditions
above water saturation where full droplet activation prior to freezing cannot be guaranteed.

In the deterministic concept (Langham and Mason, 1958) ice nucleation is assumed to take place at so-called ice nucleation active sites on the particle's surface (Vali, 1966). The probability of such a site to be present on a particle and thus of the particle to nucleate ice at a certain temperature scales with the particle's surface area (Archuleta et al., 2005; Connolly et al., 2009; Welti et al., 2009). To account for this dependency and compare the ice nucleation ability of the different dust samples
the ice-active surface site density $n_{\mathrm{s}}$ was calculated:

$$n_{\mathrm{s}} = -\frac{\ln(1-AF)}{\overline{A_{\mathrm{ve,w}}}} \approx \frac{INP}{N_{\mathrm{tot}}\overline{A_{\mathrm{ve,w}}}} \tag{1}$$

with the total particle concentration $N_{\mathrm{tot}}$ and the activated fraction $AF$ given by $INP/N_{\mathrm{tot}}$. The approximation is only valid for $AF < 0.1$ which is the case in this study. As aerosol particles larger than 1 $\mu$m were in the size range of the ice crystals and could not be differentiated based on size in the optical particle counter spectra, the $n_{\mathrm{s}}$ was corrected by subtracting the average
$n_{\mathrm{s}}$ at $RH_{\mathrm{i}}$ = 100 to 103%. At these low $RH_{\mathrm{i}}$ values no ice nucleation is expected, thus counts in the size range of ice crystals are assumed to be large dust particles.

The ice nucleation activity of the Australia, Atacama milled, Etosha, Tenerife2014_1, Peloponnese and Morocco samples was additionally measured after they had been exposed to 300°C for 10 h. For these experiments the tank was not used. Instead, dust was dry dispersed using particle-free air from a sonicated flask via a cyclone with a cut-off of 2.5 $\mu$m and a diffusion dryer into PINC, a CPC, an APS, and a SMPS. The unheated samples were additionally measured with the same set-up to allow direct comparison. The $n_{\mathrm{s}}$ measured with the tank set-up in comparison to the sonicated flask set-up for the unheated

samples varied between a factor of 1/3 to 3, which is a good agreement given the limitations of $n_s$ as a comparison parameter (Hiranuma et al., 2015b, 2018). Possible reasons for the differences are the sonicated flask breaking up larger entities, thus leading to a different mineralogy per size bin than the rotating brush generator, or uncertainties stemming from the use of a different measurement set-up. This shows the limits of using $n_s$ for such complex, polydisperse samples, which ideally should remove any size dependency. To reduce set-up dependent uncertainty, only results measured with the same measurement set-up are compared in the following discussion.

## 3   Results and Discussion

### 3.1   Ice nucleation in the deposition and condensation mode and dust mineralogy

Ice-active surface site density was determined for 18 dust samples of which 4 are from the Sahara and were collected after atmospheric transport at the Izaña observatory in Tenerife, and 3 after atmospheric transport in the Peloponnese, in Crete and the Sinai Peninsula in Egypt. Figure 1 shows RH-scans at three temperatures for these Saharan samples together with RH-scans measured online at the Izaña observatory during the CALIMA campaigns in August 2013 and 2014, when the observatory was located in the Saharan Air Layer. The data and description of the location and campaigns are given in Boose et al. (2016b). The online and offline measured $n_s$ agree well. Most of the $n_s$ curves of the different Saharan samples span an order of magnitude. In Fig.2 a scatter plot of $\ln(n_s)$ at $RH_w = 102\%$ against the quartz plus feldspar content of the dust samples is shown which is discussed in more details below. It reveals that the airborne Saharan samples are similar in $n_s$ and their quartz plus feldspar fraction (20 - 30 wt%). This holds true for most other mineral components, as shown in Table 1. The biggest differences are found for smectite and calcite, the latter accounting for 25 to 33 wt% in the Crete, Egypt and Peloponnese samples but only for 4 to 7 wt% in the Tenerife samples. Smectite on the other hand was only found in the Tenerife samples (23 to 32 wt%) and not in dust from the other three locations. Smectite and calcite are both known to have a low ice nucleation activity (Atkinson et al., 2013; Kaufmann et al., 2016). Thus, the similar amount of one or the other mineral in all Saharan dust samples is in line with the observed similar $n_s$. Differentiating between microcline and orthoclase and comparing the bulk mineralogy to the surface-dependent ice nucleation ability, introduces uncertainty. Within this uncertainty, there is no detectable effect from the presence of microcline versus orthoclase in the airborne samples at the studied temperatures. This is in line with the findings by Whale et al. (2017) that there is no correlation between ice nucleating ability and the level of ordering in the aluminosilicate framework, which determines if orthoclase (less ordered) or microcline (more ordered) is present.

To determine how well the dust mineralogy can overall predict the ice nucleation activity, the correlation of $\ln(n_s)$ with the fractions of the most common minerals in the dust samples was compared in the partner paper (Boose et al., 2016c). This comparison showed that the immersion mode ice nucleation activity correlates best with the K-feldspar fraction alone at $T = 253$ K, a temperature where only feldspar minerals are found to have significant ice nucleation activity (Atkinson et al., 2013). At $T \leq 245$ K, the ice nucleation activity correlates best with the quartz and quartz+feldspar fractions of the dust samples (Boose et al., 2016c). At these lower temperatures, quartz and to a lower degree also clay minerals were found to nucleate ice efficiently (Atkinson et al., 2013). Before investigating the role of other compounds, here we do a similar correlation

analysis of the $\ln(n_s)$ at $T$ = 238, 240 and 242 K and $RH_w$ = 97% and 102% with the fraction of various minerals contained in the dust samples. Figure 2 shows exemplarily a scatter plot of $\ln(n_s)$ at $RH_w$ = 102% against the quartz plus feldspar content of the dust samples. The Etosha sample was excluded from the correlation as it does not contain any significant amount of these minerals. The surface-collected samples with a high ratio of quartz and feldspar tend to have a higher $n_s$ than the airborne samples. This trend is similar at the three investigated temperatures. The resulting correlation coefficients for the investigated mineral fractions are provided in Table 2. The findings by Boose et al. (2016c) for the immersion mode $n_s$ are confirmed in the deposition and condensation mode: at all three tested temperatures, the highest correlation of $n_s$ is found for the fraction of quartz, followed by the quartz + feldspar fraction. The correlation with quartz alone is dominated by the Australia sample, which consists of 91 wt% quartz and is by far the most ice nucleation active dust sample. The remaining samples, which consist of at most 64 wt% of quartz, correlate only weakly with quartz alone (e.g. R = 0.29, p=0.27, at $T$ = 238 K and $RH_w$ = 102%) and better with quartz plus feldspar (R = 0.41, p=0.08). Adding illite to the quartz + feldspar fraction leads to insignificant changes of the correlation coefficient while adding kaolinite reduces it. Calcite and kaolinite alone are always negatively correlated with $n_s$ in both deposition and condensation mode. These observations are reasonable when compared to the findings of Atkinson et al. (2013), who found a higher $n_s$ for quartz over clay minerals at $T > 236$ K but lower $n_s$ for quartz than for feldspars for $T > 242$ K. Generally, the correlations are lower in this study compared to the immersion mode data from Boose et al. (2016c). A possible reason for this is that the only partial activation of INPs at these measurement conditions in PINC due to an inhomogeneous RH-profile along the particle trajectories inside the chamber (Garimella et al., 2017) weakens the effect of differences in mineralogy. Another reason could be that surface coatings play a more prominent role at lower $RH$ values because they are less diluted than in immersion mode. Correlating mineralogy, which is based on the bulk fraction, with the surface property $n_s$, leads to additional uncertainty, as described above. Effects by non-mineral substances such as coatings are discussed in the following.

## 3.2 Ice nucleation and heat labile material

In this section, the role of heat labile material on the surface of dust particles is investigated. A representative subset of the samples was selected to reduce the number of experiments necessary. The Australia and Morocco sample were selected because of their exceptional high $n_s$, the Etosha sample, because its mineralogy did not explain the observed $n_s$, the Atacama milled sample because we expected a higher $n_s$ from the mineralogy, and finally the Tenerife2014_1 and Peloponnese samples as representatives for two airborne samples from different locations. Figure 3 shows the $n_s$ at 240 K and 242 K of the unheated and the heated samples. In case of the Tenerife2014_1 sample the heat treatment led to an increase of $n_s$ at $RH_w < 100$ % while it had no effect above water saturation. In contrast, the maximum $n_s$ of the Etosha sample decreased by one order of magnitude at 240 K after heating. At 242 K the $n_s$ was below the detection limit as can be seen by a scattered flat $n_s$ curve in Fig. 3b. The heating had a small decreasing effect on the $n_s$ of the Peloponnese sample at all $RH$ values and little to no significant effect on the $n_s$ of the remaining samples. A change in ice nucleation ability due to the heating gives an indication on the nature of the active sites, i.e. if they are inherent to the minerals themselves or a semi-volatile coating material or due

to biological particles, which are impacted by the heating. An $RH$ dependency of the change in ice nucleation ability may suggest that the material which contains or coats the active sites is susceptible to dissolution.

We investigated these possible implications further by thermogravimetric analysis, IR-ATR and Raman spectroscopy of the samples. The relative mass loss under increasing temperature is shown in Fig. 4a and its first derivative in Fig. 4b. If pure samples of single minerals or organic species were studied, the TGA would show discrete steps in mass loss, indicated by

spikes in the derivative plot. However, the complexity of the dust samples in this study, which consist of several minerals and likely also various other components, cause the spikes to widen and hardly discrete steps to be observed. For example, Xi et al. (2004) showed that the mixture of a montmorillonite with varying concentrations of organic surfactant shifts the peak in the derivative mass between the higher temperature of the montmorillonite to the lower temperature of the surfactant, depending on the concentration of the surfactant. This makes it impossible to identify exactly which species are evaporated at which

temperature. However, taking into account the temperature range, a qualitative analysis is possible.

The Morocco, Australia and Atacama milled samples, which all showed no change in $n_s$ after heating, exhibit a small, gradual decrease in mass of at maximum 0.5% between 40°C and 300°C. In the case of the three samples whose $n_s$ changed after heating, i.e. the Etosha, Peloponnese and Tenerife2014_1 samples, a larger mass loss is found. A first decrease in mass of about 1% (Etosha) to 2% (both airborne samples) is observed between 40°C and 110°C. The Etosha sample shows a second

graduate mass release from 220°C onwards (1%). The two airborne samples Peloponnese and Tenerife2014_1 show small steps in mass release between 110°C and 225°C and 110°C and 170°C, respectively, and a continuous reduction in mass above these temperatures, reaching a total loss of 3% and 8% at 300°C, respectively.

The temperature ranges where the mass loss occurs can be related to different materials which were evaporated and potentially altered the ice nucleation behavior. The first decrease in mass at 40°C to 110°C is mostly due to the evaporation of adsorbed

water on the surface of the dust particles or of volatile material such as volatile organics. At temperatures between 110°C and 300°C the mass loss is mainly related to decomposition of organic matter, e.g. amides, carboxylic and phenolic functional groups (Miyazawa et al., 2000) or the combustion of certain organic compounds such as cellulose (Stamm, 1956; Lipska and Parker, 1966). Kristensen (1990) studied various biogenic organic materials, such as cellulose, glucose, bacteria (Escherichia coli, bacillus subtilis), and humic and glutamic acid, which showed either a bimodal or trimodal pattern in the TGA derivative.

They observed for all samples a first peak at 160-180°C which accounted for 10-42% of the mass loss and a second mass loss peak between 340°C and 490°C. Most minerals present in the dust samples are stable at T $\leq$ 300°C apart from smectite and gypsum (Földvári, 2011), as further discussed below.

We performed IR-ATR spectroscopy on the Etosha and Tenerife2014_1 to investigate the nature of the material responsible for the respective decrease and increase in $n_s$ with heating. Furthermore, we chose the Australia sample as representative for

most other cases, where the $n_s$ stayed the same. Figure 5 shows the IR-ATR spectra of the three samples before and after heating. The bands between wavenumber 700 to 1200 cm$^{-1}$ are related to the dominant minerals in the (bulk) samples: quartz in case of the Australia sample; dolomite, calcite and ankerite in the case of Etosha; and kaolinite and smectite in case of the Tenerife2014_1 sample. Kaolinite also has signals at 3619, 3650 and 3685 cm$^{-1}$ (Lafuente et al., 2015). No significant differences are observed between the unheated and the heated Australia and Etosha samples. The intensity differences in the

mineral bands are likely related to single grains unrepresentatively amplifying the signal during sampling. The Tenerife2014_1 sample on the other hand shows clear differences before and after heating. A loss of intensity in the OH stretch region between 3500 and 3100 cm$^{-1}$ is observed, which is too pronounced to be from adsorbed water only. Further intensity loss is observed in the C-H aliphatic region with decreasing bands between 3000 and 2850 cm$^{-1}$. This points to volatile organics being present on the unheated sample which were released during heating.

Raman mapping was performed on the Etosha, Tenerife2014_1 and Australia samples. Due to strong fluorescence, however, the Tenerife2014_1 spectra did not yield any information and are thus not presented here. The Raman maps for the Etosha and Australia samples are shown in Fig. 6. They reveal bands between 1200 and 1700 cm$^{-1}$ with a distinctive pattern related to soot and carbonaceous material (Sadezky et al., 2005) being present on many particles in the Australia sample (Fig. 6a-2 and 6a-4) and on some in the Etosha sample (Fig. 6c-2 and 6c-4). The carbonaceous material is not affected by the heating (Fig. 6b and 6d). A cluster with a strong broad signal at 3180 cm$^{-1}$ and a secondary band at around 1080 cm$^{-1}$ is observed for the unheated Etosha sample, which is present on most particles (Fig. 6c-1 and 6c-3). After the heat treatment, no cluster is identified anymore containing the broad 3180 cm$^{-1}$ band, yet the signal at 1080 cm$^{-1}$ remains. This supports the interpretation that the band at 1080 cm$^{-1}$ can be attributed to calcite. Furthermore, the band at 1080 cm$^{-1}$ does not correlate with the 3180 cm$^{-1}$ band, therefore they belong to two different compounds. The cluster analysis groups these signals for the unheated sample due to the spatial proximity of the materials on the dust. It is likely that on mineral dust grains with a high calcite content a compound was absorbed. During the heat treatment this absorbed compound disappeared and the calcite remained. Afterwards, the cluster analysis showed no group containing the 3180 cm$^{-1}$ band. A filter (3180±175 cm$^{-1}$) was applied to search specifically for this signal. This revealed only very few particles carrying the related material after heating (Fig. 6d-1 and 6d-3).

Identification of the material which was released or decomposed during the heating was hampered by fluorescence inherent to the minerals in the samples and also possibly due to biological material if present in the unheated samples. The signal to noise/fluorescence ratio was optimized by impacting small amounts of the samples on a pure aluminum surface and by adjusting the laser power but the fluorescence could not be entirely suppressed. This, together with the complexity of the samples, inhibited an unambiguous identification of the species which were altered by the heating and may affect the ice nucleation ability. We suggest three possible candidates for the cluster with a strong Raman signal at 3180 cm$^{-1}$ in the Etosha sample: a) Amides typically show a Raman signal between 3300 and 3100 cm$^{-1}$ (Socrates, 2001) as observed in the Etosha sample but distinct peaks are usually also observed for amides between 1700 and 1600 cm$^{-1}$, which are absent in the Etosha sample. b) Pure ammonium sulfate has a broad band above 3000 cm$^{-1}$ and a sharp peak at around 990 cm$^{-1}$ (Bertram et al., 2011). While the broad band at 3180 cm$^{-1}$ agrees well with the observed signal, a second band is found at 1080 cm$^{-1}$ in the Etosha sample. Ammonium sulfate has a band at 990 cm$^{-1}$ which is significantly different from 1080 cm$^{-1}$. Thus, the signal can be attributed to the calcite which has a very prominent band at this position. In addition, the XRD measurements confirmed calcite to be present in the sample. c) Cellulose shows a broad band between 3575 and 3125 cm$^{-1}$ and numerous bands between 1320 and 1030 cm$^{-1}$ (Socrates, 2001). Additional bands at 1750 and 1725 cm$^{-1}$, 1635 and 1600 cm$^{-1}$ and 1480 and 1435 cm$^{-1}$ are not distinguishable in the Etosha Raman spectrum.

For the Etosha sample an effect of organic or other heat labile material on the ice nucleation behavior appears likely. The main minerals contained in the Etosha sample (i.e. ankerite, calcite, dolomite, and muscovite) are not known to be particularly ice nucleation active at the investigated temperatures. In case of ankerite the ice nucleation ability is unknown. Based on its similarity with dolomite, a carbonate known not to be ice nucleation active, it is assumed that ankerite is also not active. Thus, one of the suggested candidates with the strong Raman signal at 3180 cm$^{-1}$ is likely responsible for the ice nucleation activity of the Etosha sample. Being part of proteins, amides are ubiquitous in nature. Similarly, cellulose is the most abundant organic compound on Earth (Kamide, 2005) amongst others as structural component of algae. Both cellulose (Hiranuma et al., 2015b) and some proteins (Maki et al., 1974) have been identified to cause ice nucleation at the studied temperatures. Ammonium sulfate has been observed to correlate with higher INP concentration in Saharan dust (Boose et al., 2016b) and to increase the freezing onset temperature of microcline (Kumar et al., 2018a) and various other minerals (Whale et al., 2018) by up to 3 K. According to a study on the Ntwetwe Pan in Botswana by Thomas et al. (2014), the organic carbon concentration in a salt pan is about 1 wt% at the surface and consists of cyanobacteria and algae. We assume that similar values apply also for the Etosha pan. The Etosha sample was collected from the edge of the salt pan, a few hundred meters away from a fertile soil area containing the highest organic carbon content of the national park (Beugler-Bell and Buch, 1997). As wind erosion was identified in these nearby fertile soils, aeolian transport potentially led to higher organic matter concentration at the edge of the pan compared to further to the center of the pan. Overall, this suggests that the Etosha sample's ice nucleation ability is almost entirely caused by organic matter, potentially cellulose or proteins which were mixed with or adsorbed onto the mineral dust on the ground explaining the almost complete suppression of ice nucleation of the heated samples.

In contrast to the Etosha sample, the Tenerife2014_1 sample consists of a number of minerals ice nucleation active at the studied temperatures, e.g. orthoclase, plagioclase, and quartz (Table 1). The sample shows the largest mass loss in the TGA analysis. The rather steep step in the TGA loss curve at about 120°C suggests a certain species to be released at this temperature, probably containing aliphatic compounds as suggested by the IR-ATR measurements. The complexity of the airborne Saharan samples is indicated by García et al. (2017), who studied organic material in the Saharan Air Layer. They collected 42 PM2.5 and PM10 filters at the Izaña observatory in parallel to the Tenerife2013 sample collection and the CALIMA2013 campaign in August 2013. Organic matter accounted for about 1.5 wt% of the aerosol composition in the Saharan Air Layer and was determined to mainly consist of saccharides, related to organic material in soils, biogenic secondary organic aerosol particles resulting from isoprene and $\alpha$-pinene oxidation, and organic compounds from natural and anthropogenic sources such as vegetation and engine emissions. During daytime, the boundary layer reaches the altitude of the Izaña observatory and organic matter can originate from local sources. During nighttime the observatory is located in the free troposphere and aerosol sources are distant. Anthropogenic and natural emissions can originate from the North African coast or be advected from Europe (Rodríguez et al., 2011; García et al., 2017). As the dust collection took place over several days and nights, an influence of organic matter from local sources cannot be excluded.

Another explanation for the reduction in mass at temperatures below 300°C is the release of free water molecules from the crystal lattice as indicated also in the IR-ATR spectra between 3600 and 3100 cm$^{-1}$. In some cases, this affects the crystal lattice: Smectite, a swelling mineral, collapses under decreasing water vapor pressure as experienced during the heating. This

decreases the interlayer spacing (for an overview of the effect of layer charge on smectite swelling see Laird (2006)). It is unknown if the change in crystal lattice has an effect on the ice nucleation ability of the otherwise only weakly ice nucleation active smectite (Pinti et al., 2012; Kaufmann et al., 2016). An effect cannot be excluded as lattice match with ice is believed to be one of the factors promoting ice nucleation (Pruppacher and Klett, 1997). In the studied samples, smectite was present in the Tenerife2014_1 sample (23 wt%) and traces were found in the Etosha sample (1 wt%). Thus, the collapse of the smectite

lattice should only influence the Tenerife2014_1 sample. In case it has an influence, this would be related to an increase in ice nucleation ability.

XRD analysis of the unheated and heated Tenerife2014_1 sample show the conversion of gypsum to anhydrite (Fig. 7). Gypsum has a low ice nucleation ability, similar to the clay minerals kaolinite and illite (Zimmermann et al., 2008). Grawe et al. (2018) found anhydrite to have a higher ice nucleation activity than quartz in the immersion mode at temperatures below 243 K

when dry generated but a much lower $n_s$ when particles were generated from an aqueous solution. Anhydrite transforms back to gypsum when exposed to a relative humidity higher than 97% at room temperature (Bracconi et al., 2010). However, this process occurs on the order of hours to days, in line with the observed differences in $n_s$ between wet and dry generated particles in Grawe et al. (2018). Potentially, the transformation to anhydrite during heating explains the higher $n_s$ of the heated Tenerife2014_1 sample compared to the unheated one at subsaturated conditions. In this case, a partial conversion of anhydrite

back to gypsum during RH conditions above water saturation might explain the unchanged $n_s$ of the unheated and heated Tenerife2014_1 sample above water saturation. It should be kept in mind that the bulk mineralogy as determined by XRD is not necessarily representative for the particle surface where ice nucleation takes place. As needle formation has been observed in the transformation of gypsum to anhydrite (Azimi and Papangelakis, 2011; Grawe et al., 2016), we use the occurrence of needles in our sample as an indication that the gypsum - anhydrite transformation took place on the surface of particles and thus

might be responsible for the change in ice nucleation behavior of the sample. In SEM images of the unheated Tenerife2014_1 sample (Fig. 8a) hardly any needles are visible. A small number of needles is observed at the center of the image of the heated sample (Fig. 8b) while no needles are found in other SEM images of the heated sample (see the supplementary material). The apparently limited needle formation and the fact that only about 1 wt% gypsum is contained in the sample, suggests that gypsum transformation under heat treatment should only have a small effect on the ice nucleation behavior of the Tenerife2014_1

sample. However, given that at maximum only about 10 % of the particles act as INPs in case of the Tenerife2014_1 sample, the gypsum-anhydrite transformation might be non-negligible. Additionally, we suggest that the increase in $n_s$ found under subsaturated $RH_w$ conditions for the Tenerife2014_1 sample is caused by the volatilization of aliphatic compounds containing matter, as indicated by the IR-ATR and TGA measurements, which inhibited the active sites of the mineral dust itself.

**4   Conclusions**

In this study we showed that the fractions of quartz and the sum of quartz and feldspars in desert dust samples correlate better than all other mineral fractions with the ice nucleation active surface site density of the dust in deposition and condensation

mode at temperatures between 238 K and 242 K. This is in line with the observations for the immersion mode presented in part 1 (Boose et al., 2016c) of this study. The high abundance of quartz in soils worldwide, its resistance to chemical weathering

processes and its presence in particles of all sizes make it a potentially widely spread atmospheric INP type. According to a recent study by Kumar et al. (2018b), the variation in quartz ice nucleation ability found in laboratory studies (Atkinson et al., 2013; Zolles et al., 2015; Kaufmann et al., 2016) and the superior ice nucleation ability of the quartz-rich samples from Australia and Morocco in this study and its partner paper, may be explained by the pre-processing of the samples. Milling of quartz samples, as done in our study, increases the ice nucleation ability of quartz by creating Si-O• and Si• radical sites

which can then react with water vapor (Kumar et al., 2018b). However, milling may not be the only reason for formation of the silanol (Si-OH) groups on the surface of quartz, because exposure to water molecules in ambient humidity could also result in passively converting surface siloxane groups (Si-O-Si) to silanol groups (Boehm, 1966; Wang et al., 2018). As such quartz samples may still exhibit high ice nucleation activity in the absence of milling due to the particles chemical history. Thus, it remains an open question, how much quartz contributes to the ice nucleation ability of (unmilled) atmospheric dust.

Apart from mineralogy, the ice nucleation activity of desert dust is found to be influenced by organic material mixed with the dust. In a carbonaceous sample from the Etosha salt pan, where less than 1 wt% quartz and no feldspar were present, the ice-active surface site density is found to be almost entirely due to organic matter, likely cellulose or proteins, which are mixed with the dust. On the other hand, the deposition mode ice nucleation activity of an airborne Saharan dust sample was found to increase after heating. Three potential explanations are found, two of them related to changes in the mineralogy: While it

cannot be excluded that the increase in $n_s$ was caused by a change in lattice spacing due to interlayer water release, it seems more likely that gypsum transforming to anhydrite made the sample more ice nucleation active. The Tenerife2014_1 sample is the only gypsum-containing sample that was investigated after heating, thus it remains an open question if and how much anhydrite contributed to the increase in $n_s$. Another reason for the increase could be that the ice nucleation active sites of the unheated sample were blocked by volatile organic material. The volatilization of the aliphatic compounds during the heating

recovered these active sites. This is further supported by the observation that no difference was found for the same sample in the condensation freezing mode, indicating that the water condensing on the surface may also recover the active sites. As this change in ice-active surface site density was not observed for a second airborne Saharan dust sample, it suggests that different aging processes or mixing of Saharan dust with organic material during atmospheric transport can influence the dust's ice nucleation ability in both directions.

*Data availability.* Ice nucleation data from this study are available here: Boose et al. (2019). Additional SEM images can be found in the supplementary. Additional data are available upon request.

*Author contributions.* YB collected the Tenerife and Israel samples, conceived, and lead the measurement campaign, performed the ice nucleation measurements and analysis, performed and analyzed the XRD measurements, analyzed the TGA measurements and wrote the

manuscript, PB performed and analyzed the SEM, IR-ATR and Raman measurements and contributed to the manuscript, JO contributed to the Raman measurements, HG supervised the IR-ATR and Raman measurements and analysis and contributed to the manuscript, MP performed and analyzed the XRD measurements, ZAK performed the TGA measurements, BS, ZAK and UL supervised the project and contributed to the manuscript. All authors contributed to the interpretation of data.

*Competing interests.* The authors declare no competing interests.

*Acknowledgements.* We thank the two anonymous reviewers for their helpful comments. The various dust samples in this paper have been collected by a number of people who the authors are very thankful to: Maria Kanakidou and her team (Peloponnese, Crete), Felix Lüönd (Atacama), Paolo D'Odorico and Christopher Hoyle (Etosha), Lukas Kaufmann, Konrad Kandler and Lother Schütz (Taklamakan), André Welti (Australia, Mojave), Monika Kohn (Dubai), Joel Corbin (Morocco), Sergio Rodríguez (Tenerife) and Hamza Mohamed Hamza (Egypt). The authors would like to thank Dr. Joanna Wong for her assistance with the TGA measurements and the Laboratory of Composite Materials and Adaptive Structures, ETH Zurich for the use of their thermal analysis equipment. We thank Hannes Wydler for his technical support with PINC. PB would like to thank Karin Wieland and Rita Wiesinger for helpful discussions concerning the interpretation of the Raman spectra. HG and PB would like to thank the Analytical Instrumentation Center and the X-Ray Center of the TU Wien for the use of the Raman and XRD equipment and the University Service Centre for Transmission Electron Microscopy of the TU Wien for recording the SEM images in this work. YB and ZAK gratefully acknowledge support by the Swiss National Science Foundation (grant 200020 150169/1). The research leading to these results has received funding from the European Union's Seventh Framework Programme (FP7/2007-797 2013) under grant agreement No 603445 (BACCHUS). HG and PB gratefully acknowledge support by the FFG (Austrian Research Promotion Agency) for funding under Project No. 850689.

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

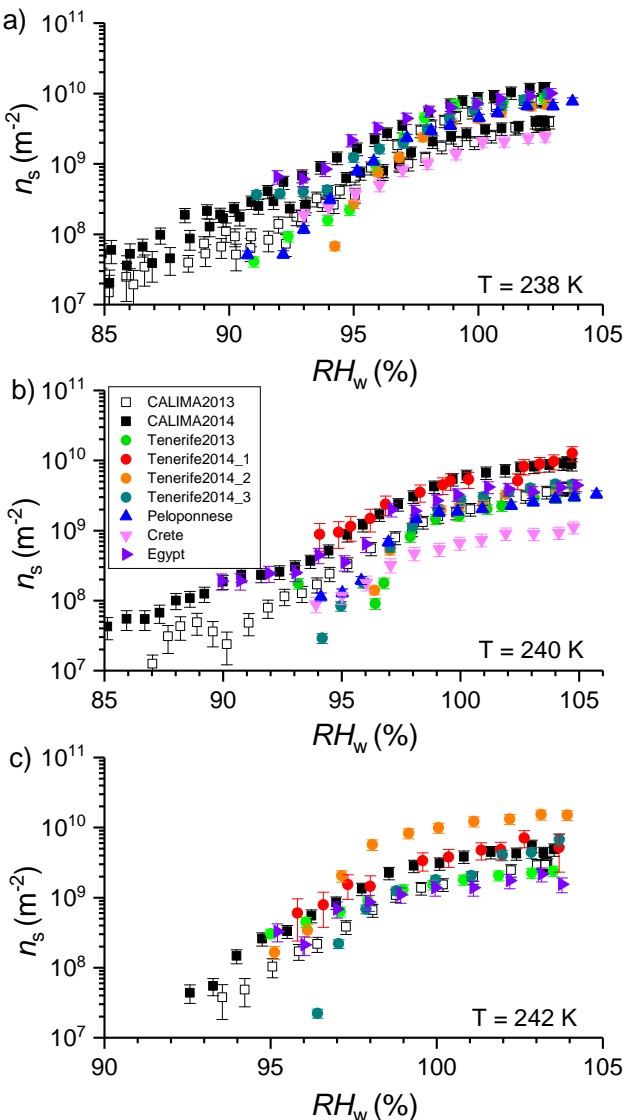

**Figure 1.** Ice-active surface site density at a) 238 K, b) 240 K and c) 242 K of samples collected airborne at the Izaña observatory on Tenerife in 2013/2014 (circles), in Egypt, Crete and Peloponnese (triangles) and measured in-situ during the CALIMA 2013 and 2014 campaigns (open and filled squares) which took place at the Izaña observatory.

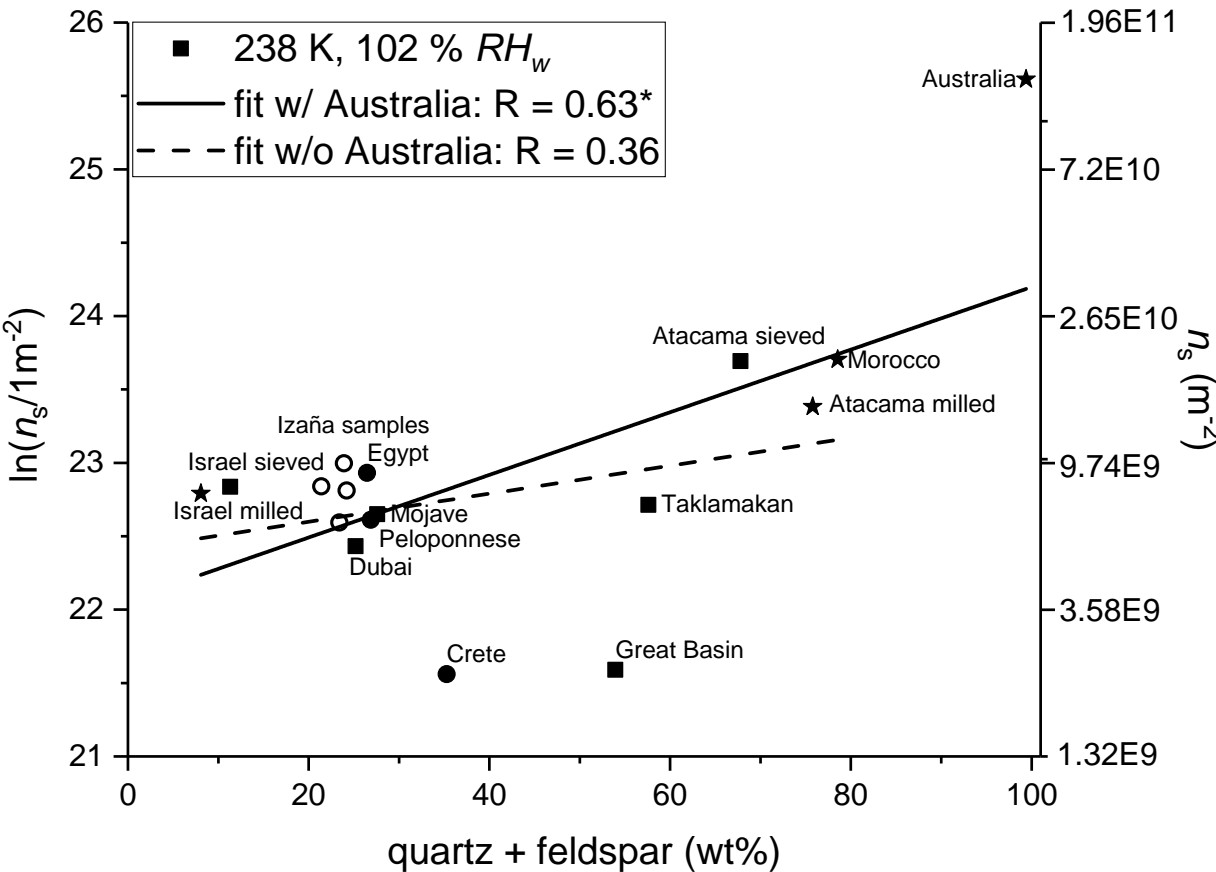

**Figure 2.** Natural log of the ice-active surface site density as a function of the sum of quartz and feldspar content of the samples. Square symbols indicate surface-collected samples, stars milled samples, and circles indicate airborne samples. For clarity, the Tenerife samples are not named individually and instead shown as open circles. The asterix in the legend indicates that the correlation is significant at the 0.05 level.

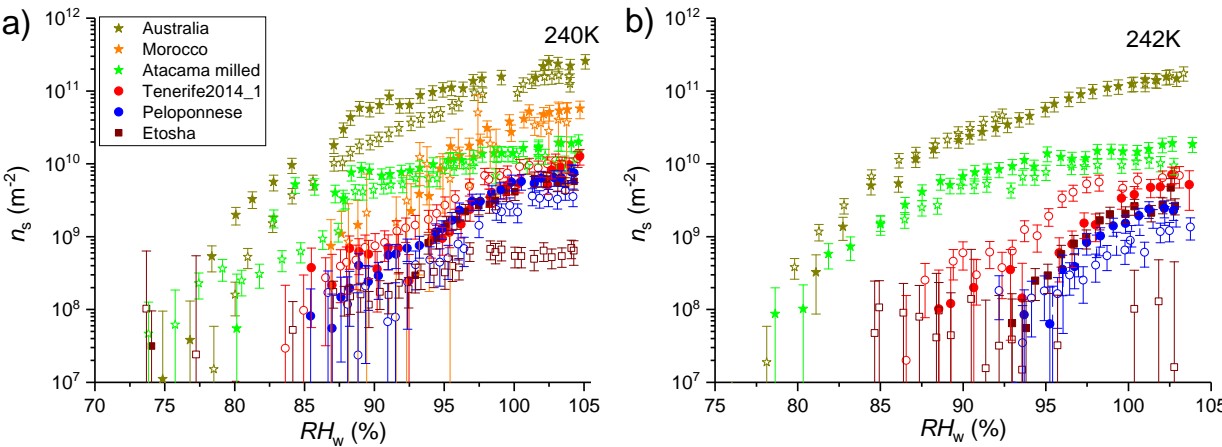

**Figure 3.** Ice-active surface site density at a) 240 K and b) 242 K of samples before (filled symbols) and after (open symbols) heat treatment. Error bars include the Poisson error of the INP measurements and the maximum variation of $\overline{A_{ve,w}}$. Square symbols indicate surface-collected samples, stars milled samples, and circles indicate airborne samples.

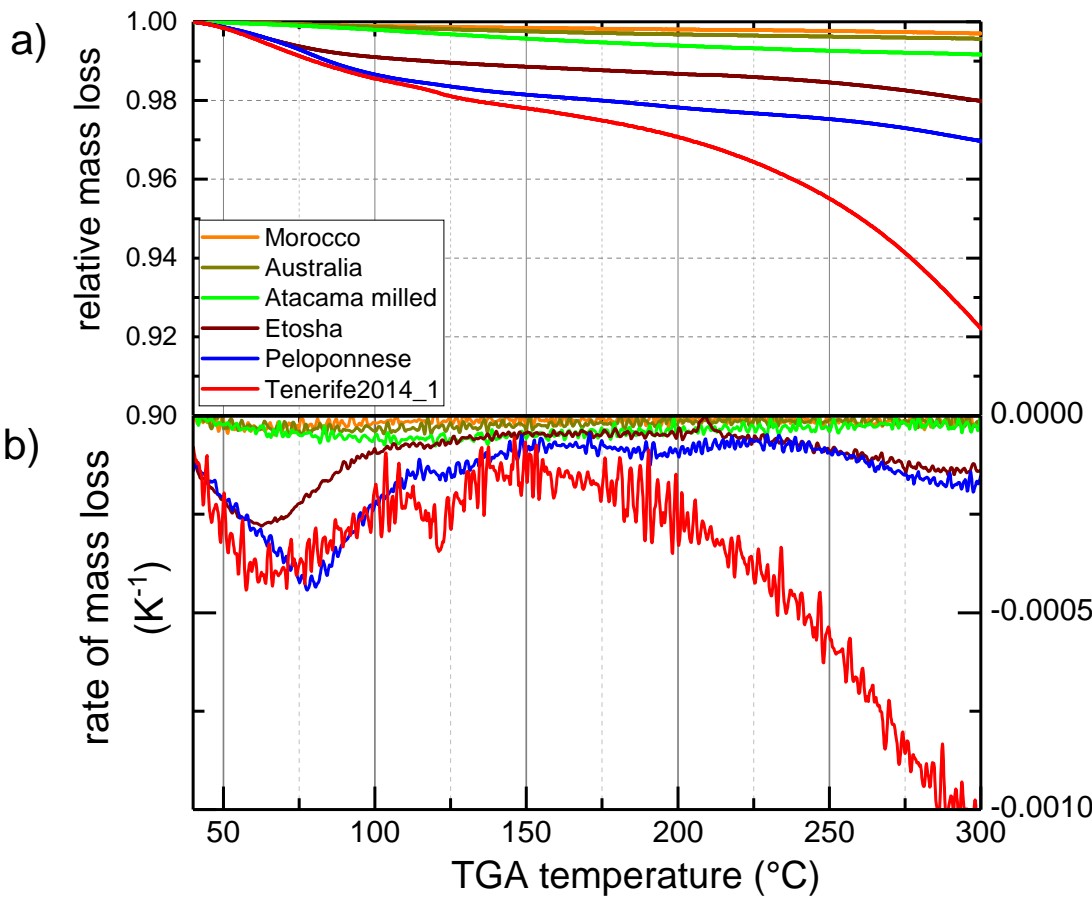

**Figure 4.** Relative mass loss (a) and its derivative (b) under heating of the different dust samples.

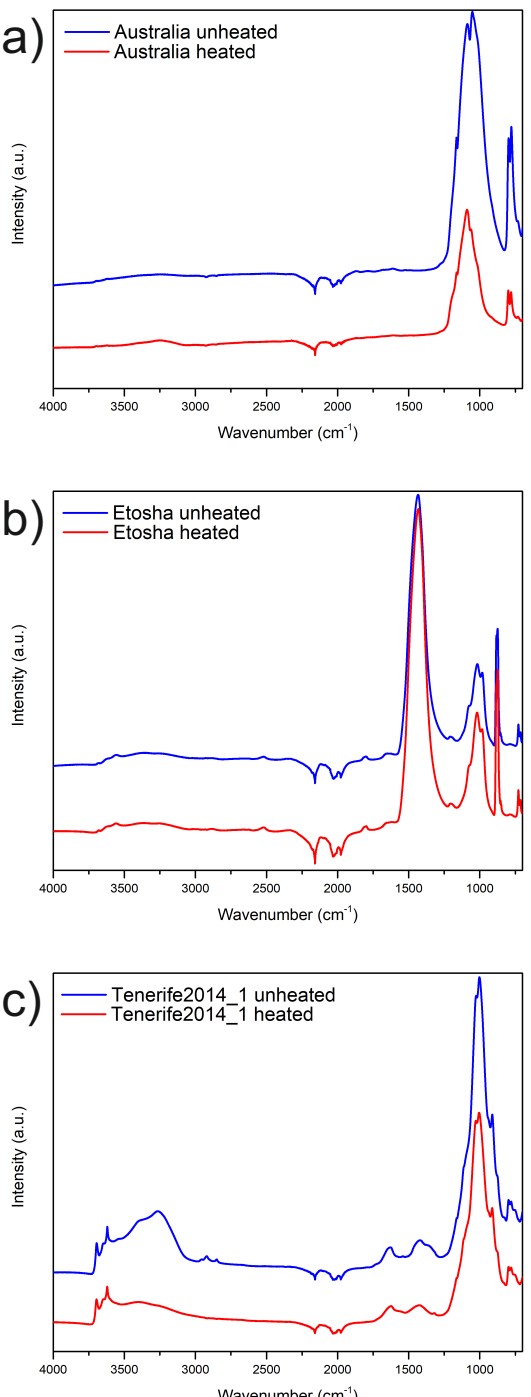

**Figure 5.** Attenuated Total Reflectance Infrared (ATR-IR) spectra for the a) Australia, b) Etosha, and c) Tenerife2014_1 samples before (blue) and after heating (red).

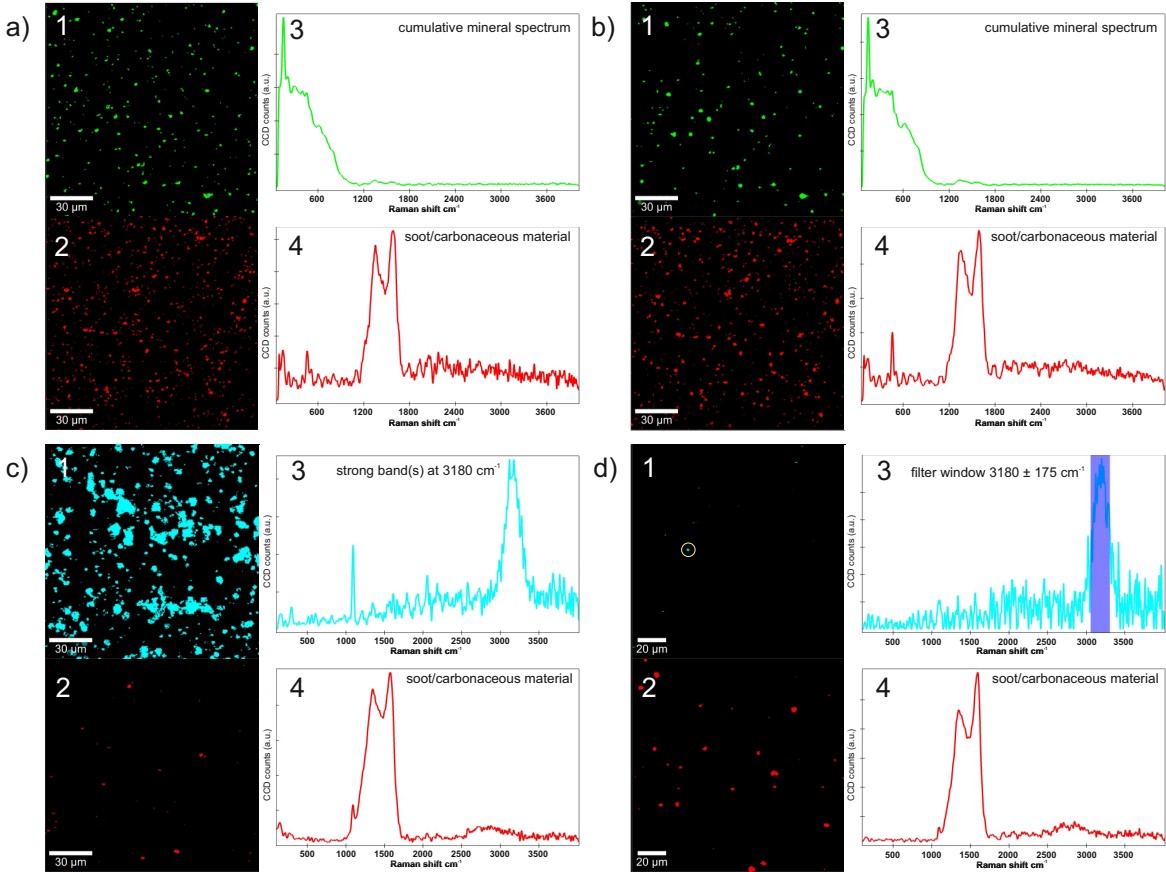

**Figure 6.** Raman mapping results for the Australia (a and b) and the Etosha sample (c and d). 1) and 2) of each panel show the location of particles from clusters with spectra similar to those shown in 3) and 4). In d1) the particle is encircled for which the filter (d3) found a spectrum.

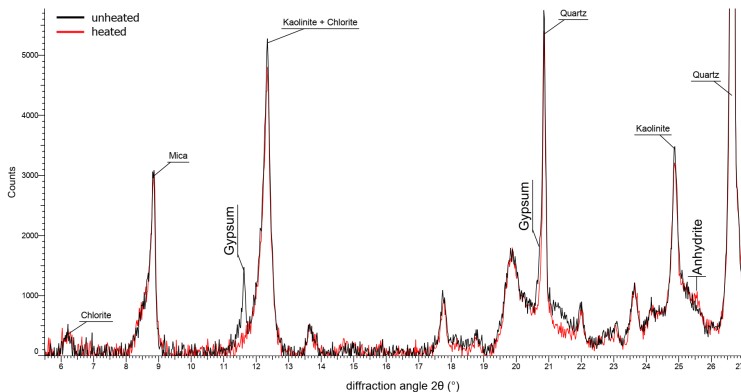

**Figure 7.** XRD diffractogram of unheated and heated Tenerife2014_1 sample. Vertical text indicates changes in peak height which were used to identify the decrease in gypsum and increase in anhydrite with heating. Horizontal text indicates peaks associated with other minerals in the sample.

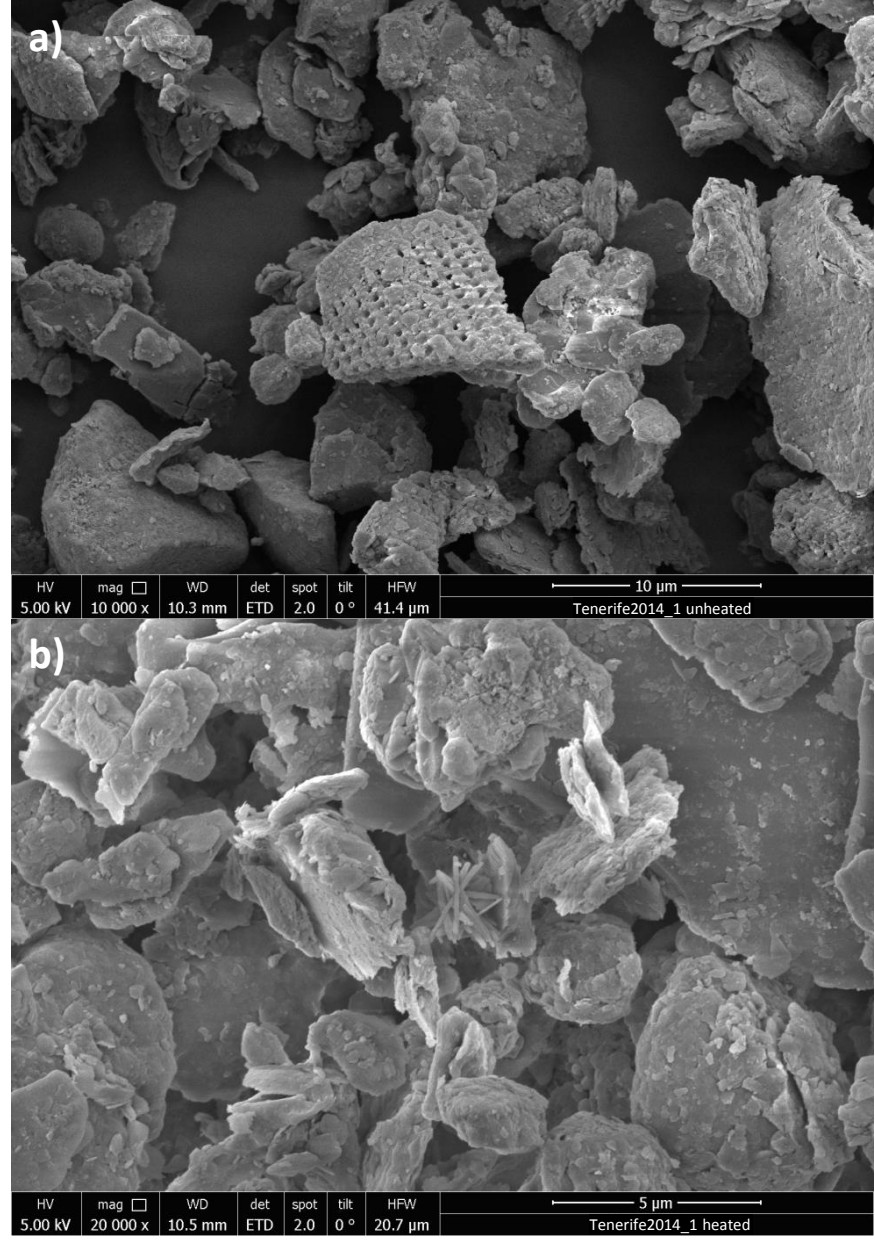

**Figure 8.** Scanning electron microscopy images of the a) unheated and b) heated Tenerife2014_1 sample. Note the different scales in the two images.

**Table 1.** Mineralogical composition in wt% of airborne Saharan dust samples. Crete, Egypt, Tenerife2013 (Tenerife in Part 1), and Peloponnese as in Boose et al. (2016c). Results were rounded to the nearest integer from the original Rietveld fit results, thus total composition $\neq$ 100 wt% may occur.

| Mineral | Crete | Egypt | Tenerife2013 | Tenerife2014_1 | Tenerife2014_2 | Tenerife2014_3 | Peloponnese |
|---|---|---|---|---|---|---|---|
| Calcite | 25 | 29 | 7 | 4 | 6 | 6 | 33 |
| Chlorite | 4 | 8 | 2 | 3 | 2 | 1 | 3 |
| Dolomite | 3 | 8 | 2 | 2 | | 2 | 5 |
| Gypsum | 4 | 6 | 2 | 1 | 3 | 2 | |
| Halite | 1 | 4 | | | 2 | 1 | |
| Hematite | 1 | | 1 | 2 | 1 | 1 | 1 |
| Illite | | | 6 | 16 | 7 | 9 | 13 |
| Kaolinite | 12 | 11 | 16 | 18 | 13 | 14 | 8 |
| Microcline | | | 4 | | 5 | 5 | |
| Muscovite | 9 | 8 | 7 | 7 | 7 | 8 | 5 |
| Orthoclase | 5 | 4 | | 4 | | | 4 |
| Palygorskite | 5 | | 2 | 2 | 3 | 3 | 5 |
| Plagioclase | 7 | | 4 | 8 | 6 | 5 | 5 |
| Smectite | | | 32 | 23 | 32 | 30 | |
| Quartz | 23 | 23 | 14 | 12 | 13 | 14 | 18 |

**Table 2.** Overview of the Pearson correlation coefficients of the sum of selected mineral fractions and $\ln(n_s)$ at different temperatures. An asterix indicates that the correlation was significant at the 0.05 level. K-feldspar comprises microcline and orthoclase, while feldspar refers to the sum of microcline, orthoclase and plagioclase. The number of samples included in each correlation varies because the $n_s$ of the Mojave, Peloponnese and Tenerife2014_2 samples was below the detection limit at 242 K, and the size distribution measurements of the Tenerife2014_1 sample were corrupted for the RH-scan at 238 K.

| $T$ | 238 K | 240 K | 242 K | 238 K | 240 K | 242 K |
|---|---|---|---|---|---|---|
| $RH_w$ | 97 % | 97 % | 97 % | 102 % | 102 % | 102 % |
| number of samples | 18 | 19 | 16 | 18 | 19 | 16 |
| K-feldspar | 0.02 | 0.03 | -0.06 | -0.08 | -0.07 | -0.13 |
| feldspar | 0.15 | 0.08 | 0.05 | 0.05 | -0.09 | -0.17 |
| quartz | 0.63* | 0.52* | 0.47 | 0.71* | 0.74* | 0.71* |
| illite | -0.14 | -0.15 | 0.11 | -0.01 | -0.04 | 0.13 |
| kaolinite | -0.66* | -0.56* | -0.41 | -0.51* | -0.35 | -0.19 |
| feldspars + quartz | 0.64* | 0.5* | 0.44 | 0.63* | 0.56* | 0.49 |
| feldspars + quartz+ illite | 0.63* | 0.49* | 0.47 | 0.65* | 0.58* | 0.53* |
| feldspars+quartz+kaolinite | 0.51* | 0.40 | 0.35 | 0.54* | 0.51* | 0.48 |
| feldspars+quartz+illite+kaolinite | 0.50* | 0.38 | 0.37 | 0.55* | 0.51* | 0.51 |
| calcite | -0.12 | 0.05 | -0.22 | -0.31 | -0.26 | -0.32 |