# Peer review of "Heterogeneous ice nucleation on dust particles sourced from 9 deserts worldwide - Part 2: Deposition nucleation and condensation freezing"

_Atmospheric Chemistry and Physics, 2018_

## Referee Comment (RC1) · Anonymous Referee #1 · 23 Nov 2018

General comments:

The present manuscript is a follow up paper to "Heterogeneous ice nucleation on dust particles sourced from nine deserts worldwide – part 1: Immersion freezing" (Boose et al, 2016). In part 2 (the present paper), ice nucleation efficiency of minerals in deposition nucleation and condensation freezing modes is studied, as well as the effect of coatings (biological and volatile/semi volatile organic material). The paper is generally well written, and it complements part one nicely. However, in some way it is difficult to follow the paper as it at times rely on the reader to remember details of part 1. Generally, the authors are asked to check that the paper is consistent with the partner paper in the use of sample names, e.g. "Tenerife" in part 1, and "Izaña" in part 2, as well as minerals (e.g. specification of Feldspar and K-Feldspar). Additionally, there is little information on why some samples are selected for further analysis and others not, e.g. XRD of one sample before and after heating, Raman mapping of two desert samples and no airborne sample, the difference in number of samples in table 2, and why is Izaña2014_2 selected and not the other Izaña samples?

In their results, the mineral fraction from feldspars and quartz correlate with the ice nucleation surface site density (ns), in both deposition and condensation mode, similar to what was found for immersion freezing in part 1. Some organic material coating the particles altered the ns, seen by comparing untreated and heated samples. In one sample, the ns is higher in the heated sample which is devoted to evaporation of volatile organic material. In this sample, also the minerology changed (gypsum to anhydrite) between the pre- and heated sample. A similar result was found in Grawe et al. (2018), but in this case it is devoted to an overestimation of ns because of large needle shaped particles that could cross the size selecting step. The authors are therefor asked to address the possibility of needle shaped particles (see more details below) and if necessary change their conclusions.

Specific comments:

Page 4 "2.1 Dust sample origins and processing": How were the samples stored for up to 8 years. Will the storage change the samples (e.g. loss of volatile compounds, change in composition due to water uptake, changes in biological material on the surface)?

Page 10 in the subchapter 3.2 "Ice nucleation and heat labile material": Three samples are discussed extensively from page 10 onwards, Etosha, Australia and Izaña 2014_2. It would be easier for the readers to have a summary of why these three samples are further investigated and discussed, compared to the rest.

We learn on page 12 (line 13) that there is a change in the mineralogical composition between untreated and heated samples. Please add the XRD results of heated and unheated samples, either to the paper or in a supplement. Large uncertainty is associated with comparing particle composition and bulk chemical analysis, which the readers also are made aware of in the paper. I would like to draw the authors attention to an article where the ice nucleation efficiency of coal fly ash particles were investigated by Grawe et al. (2018). In this case, needle shaped particles could explain the higher ns of one sample where anhydrite changed to gypsum after suspension. Many of the needles were larger than 300 nm (up to ~5 $\mu$m), but could - due to the fact that the dynamic shape factor of the needles differ significantly from unity - cross the size selection step in the DMA. Needles can be formed in both directions, from anhydrite to gypsum and gypsum to anhydrite. An example of the formation of anhydrite needles from gypsum is seen in Azimi and Papangelakis (2011). Can this also be the case of the one Izaña sample? The loss of the OH peak could also be explained by gypsum converting to anhydrite. If this is the case in your study then please change the conclusion. If not, then the discussion should cover why this volatile organic coating only applies to one sample. Isn't this expected from the other airborne samples too, at least the other Izaña samples?

Technical comments:

Title: Change '9' to 'nine' to be consistent with the partner paper.

'Ice nucleating particles' without hyphen, like in partner paper and in Vali et al. (2015).

Figure 1: The black color of the CALIMA2014 sample symbols covers all the other samples. Please change this to make it easier for the readers to see all results. Please remind the reader that CALIMA is the same sample location as Izaña in the figure legend.

Figure 2: Please add to the legend text explaining the astrix (as in table 2). Why is the Etosha sample not present in this figure?

Figure 3: Please remind the reader which samples are airborne, milled and sieved in the figure legend. Why is the Atacama sample called milled and not the Australia and Morocco sample? Why does the heated sample from Morocco only have an upper limit?

Figure 5: In the text, the samples are discussed in the following order (ref. P10, L5-14) Australia, Etosha and Izaña 2014_2. The two first show no change between unheated and heated, and in the last sample a change is discussed. It would be more logical for the reader if the locations appear in the same order as the text.

Figure 6: The figure contains a lot of information, but the grey maps (1) add no important information as the images are taken at relatively low magnification so the particles can't really be seen. Also, please remove: "see text for details".

Table 2: Please define the K—feldspar and feldspars groups in the figure legend. Please explain the readers why there are different numbers of samples.

Introduction:

Page 2, line 21: Please add a comma after the South Pole.

Page 2, line 24: The term "potassium feldspars" is used sometimes, and "K-feldspars" other times. Please be consistent.

Page 3, line 31: Please specify the minerals in the K-feldspar fraction.

Page 3, line 33: Please specify the minerals in the feldspar fraction.

Methods:

Page 4, line 23: Please add country (Crete and Peloponnese, Greece) and then use the same structure as before, e.g. "…(Crete and Peloponnese, Greece), and the 10th of May 2010 (Aburdees, Egypt)."

Page 5, line 16: "….mobility diameter between 12.2 – 615 nm…" Is the decimal place

significant?

Page 6, line 25: Change 'tank' to 'sample container'

Results and discussion:

Page 8: line 26 & 27: Please add p-value.

Page 9, line 4: Please explain why these samples are selected.

Page 10, line 6: Please remind the reader that the spectra are from bulk material. E.g. "...related to the dominant minerals (in bulk) in the samples".

Page 11, line 7 & 8: "The minerals contained in the Etosha sample..." Please remind the reader which minerals so they don't have to look it up in the partner paper.

P 11, line 9: Sentence too long.

Page 11, line 10: In this line you abruptly move from the discussion of minerals to organic material.

Page 11, line 33: Remove 'the' to During daytime,...

References:

Azimi, G., and Papangelakis, V. G.: Mechanism and kinetics of gypsum–anhydrite transformation in aqueous electrolyte solutions, Hydrometallurgy, 108, 122-129, 10.1016/j.hydromet.2011.03.007, 2011. Grawe, S., Augustin-Bauditz, S., Clemen, H. C., Ebert, M., Eriksen Hammer, S., Lubitz, J., Reicher, N., Rudich, Y., Schneider, J., Staacke, R., Stratmann, F., Welti, A., and Wex, H.: Coal fly ash: linking immersion freezing behavior and physicochemical particle properties, Atmos. Chem. Phys., 18, 13903-13923, 10.5194/acp-18-13903-2018, 2018. Vali, G., DeMott, P. J., Möhler, O., and Whale, T. F.: Technical Note: A proposal for ice nucleation terminology, Atmos. Chem. Phys., 15, 10263-10270, 10.5194/acp-15-10263-2015, 2015.

[Figure]

2018.

---

## Referee Comment (RC2) · Anonymous Referee #2 · 10 Dec 2018

Boose et al. reports measurements of the effectiveness with which a range of natural dust samples from around the world to nucleate ice from supersaturated water vapour. Various analytical techniques were used to determine the composition of the samples and to attempt to establish what components are responsible for observed ice nucleation, leading to the conclusion that quartz and feldspar content is responsible for the ice nucleation observed in most samples. In one sample organic substances were found to be largely responsible while in another removal of organics actually enhanced ice nucleation effectiveness. The conclusions of the paper are interesting and relevant

to the scope of ACP, highlighting the substantial complexity in ice nucleation by natural samples. The paper is mostly well written, sensibly structured, and is entirely suitable for publication in ACP, after a few minor corrections

Minor comments

I have a few issues with the identification of feldspar phases via powder XRD in this work. Alkali feldspar structure is a complex topic and it is not clear to me that powder diffraction is adequate to certain about which polymorph is present in dust samples of the type characterised here. Indeed in part 1 of this study the authors state 'Where microcline and orthoclase are present in the same sample, their individual fraction could not be distinguished reliably' in the caption to the relevant table. It is not clear to me what has changed in the present study. The difference between orthoclase and microcline is essentially the degree of order of the aluminosilicate framework. There is not necessarily a hard line between orthoclase and microcline. Parsons et al (American Mineralogist (2015) 100 (5-6), 1277-1303) and references therein discuss some of the relevant issues. Additionally the differences between even very pure samples of the two minerals in powder XRD are subtle. It must be more difficult to be certain of phase when mixed dusts with very low feldspar contents are under investigation. In summary, if the statements in the paper regarding the relative amounts of orthoclase and microcline are to be kept a much better description of the powder XRD procedure, and justification the differentiation between orthoclase and microcline is needed. I do not think this is particularly key to the paper, because, as the authors say, there is not any difference between the feldspars as characterised anyway. I would suggest treating the topic as it was treated in part 1 of this study.

Relatedly, Pg 8 line 12 states that microcline is the more active K-feldspar polymorph without citation. I think this statement should be changed, or its origin cited and discussed. The superior ice nucleating ability of microcline seems to be assumed throughout the paper. I would note that Whale et al. (Phys. Chem. Chem. Phys., (2017), 19, 31186) which used pure feldspars '...found no correlation between ice-nucleating efficiency and the crystal structures or the chemical compositions...' of ice nucleation active feldspars, albeit in rather different conditions.

Naming of samples is not entirely consistent through the manuscript I think, this should be checked.

Specific comments

Abstract line 7 to 9- sentence starting 'in this study,....' does not read well.

Abstract line 9- between 238 and 242K

Abstract line 15 reads poorly, missing comma after 'diminished'?

Pg 2 Line 2- delete 'already'

Pg 2 Line 25- ACP version of Paramonov et al. is available

Pg 3 line 32- clumsy wording, probably delete 'showing'

Pg 3 line 10- typo 'selected'

Pb 3 line 33- clumsy sentence structure

Pg 3 line 35- 'over' not the right work I think.

Pg 3 line 33 onwards- I note that there is interesting work on the topic of ice nucleation by quartz under review in ACPD at current (https://www.atmos-chem-phys-discuss.net/acp-2018-1020/), which may shed some light on the complexities of ice nucleation by various silicas.

Pg 4 line 7- while RH is increased

Pg 8 line 15- Some justification for using the conditions stated might aid the reader.

Pg 11 line 15- I'm not sure what is meant by the sentence 'However, it was not....'. I suggest clarifying this.

Pg 9 line 15- I am not sure what is meant by 'hardly' in this context? This sentence could be clearer.

Pg 11 line 12- remove comma after 'both'

Pg 12 line 6- brackets around '2006' after 'Laird'

Pg 12 line 10- sentence starting 'Thus, according to our observations...' is clumsy

Figure 2- It is not obvious what the * in 0.63* refers to without referring to final table.

---

## Author Comment (AC1) · 22 Dec 2018

We thank Reviewer 1 for their very constructive comments. We reproduce reviewer comments in blue in the following. Amended versions of the paper are given in *italics* for new sections and red text for the original text.

General comments:

The present manuscript is a follow up paper to "Heterogeneous ice nucleation on dust particles sourced from nine deserts worldwide – part 1: Immersion freezing" (Boose et al, 2016). In part 2 (the present paper), ice nucleation efficiency of minerals in deposition nucleation and condensation freezing modes is studied, as well as the effect of coatings (biological and volatile/semi volatile organic material). The paper is generally well written, and it complements part one nicely. However, in some way it is difficult to follow the paper as it at times rely on the reader to remember details of part 1. Generally, the authors are asked to check that the paper is consistent with the partner paper in the use of sample names, e.g. "Tenerife" in part 1, and "Izaña" in part 2, as well as minerals (e.g. specification of Feldspar and K-Feldspar). Additionally, there is little information on why some samples are selected for further analysis and others not, e.g. XRD of one sample before and after heating, Raman mapping of two desert samples and no airborne sample, the difference in number of samples in table 2, and why is Izaña2014_2 selected and not the other Izaña samples? In their results, the mineral fraction from feldspars and quartz correlate with the ice nucleation surface site density (ns), in both deposition and condensation mode, similar to what was found for immersion freezing in part 1. Some organic material coating the particles altered the ns, seen by comparing untreated and heated samples. In one sample, the ns is higher in the heated sample which is devoted to evaporation of volatile organic material. In this sample, also the minerology changed (gypsum to anhydrite) between the pre- and heated sample. A similar result was found in Grawe et al. (2018), but in this case it is devoted to an overestimation of ns because of large needle shaped particles that could cross the size selecting step. The authors are therefor asked to address the possibility of needle shaped particles (see more details below) and if necessary change their conclusions.

We thank the reviewer for these helpful comments and have addressed them below.

We have changed the naming of the Izaña201x_x samples into Tenerife201x_x and have numbered the 2014 samples from 1 to 3 to match part 1 of the paper:

Izaña2013_2 ➜ Tenerife2013
Izaña2014_2 ➜ Tenerife2014_1
Izaña2014_3 ➜ Tenerife2014_2
Izaña2014_5 ➜ Tenerife2014_3

We have also clarified the specification of components in the Feldspar and K-Feldspar group (see below).

**Specific comments:**

Page 4 "2.1 Dust sample origins and processing": How were the samples stored for up to 8 years. Will the storage change the samples (e.g. loss of volatile compounds, change in composition due to water uptake, changes in biological material on the surface)?

We have added the following part on sample storage and potential effects:

Pg.4, ln. 31: *Before arriving to the laboratory, samples were stored in various ways: Samples collected from the surface were typically stored for several weeks in PET bottles or other plastic containers. Airborne samples were stored in polypropylene tubes and sealed with paraffin wax tape. In the laboratory, all samples were stored in the dark at room temperature in polypropylene tubes after pre-processing (sieving/milling, see below). While changes in the ice nucleating ability due to water uptake, loss of volatile material or growth of biological material which may occur during storage cannot be excluded, they are assumed to be minor because the samples were collected and stored under dry conditions, hardly exposed to air and kept at a lower temperature than at which they were collected.*

Page 10 in the subchapter 3.2 "Ice nucleation and heat labile material": Three samples are discussed extensively from page 10 onwards, Etosha, Australia and Izaña 2014_2. It would be easier for the readers to have a summary of why these three samples are further investigated and discussed, compared to the rest.

We have added a sentence on

Pg. 10, ln. 28: *We performed IR-ATR spectroscopy on the Etosha and Tenerife2014_1 to investigate the nature of the material responsible for the respective decrease and increase in $n_s$ with heating. Furthermore, we chose the Australia sample as representative for most other cases, where the $n_s$ stayed the same. Figure 5 shows the IR-ATR spectra of the three samples before and after heating.*

And further on

Pg. 11, ln. 5: *Raman mapping was performed on the Etosha, Tenerife2014_1 and Australia samples. Due to strong fluorescence, however, the Tenerife2014_1 spectra did not yield any information and are thus not presented here. The Raman maps for the Etosha and Australia samples are shown in Fig. 6.*

We learn on page 12 (line 13) that there is a change in the mineralogical composition between untreated and heated samples. Please add the XRD results of heated and unheated samples, either to the paper or in a supplement. Large uncertainty is associated with comparing particle composition and bulk chemical analysis, which the readers also are made aware of in the paper. I would like to draw the authors attention to an article where the ice nucleation efficiency of coal fly ash particles were investigated by Grawe et al. (2018). In this case, needle shaped particles could explain the higher ns of one sample where anhydrite changed to gypsum after suspension. Many of the needles were larger than 300 nm (up to ~5 μm), but could - due to the fact that the dynamic shape factor of the needles differ significantly

from unity - cross the size selection step in the DMA. Needles can be formed in both directions, from anhydrite to gypsum and gypsum to anhydrite. An example of the formation of anhydrite needles from gypsum is seen in Azimi and Papangelakis (2011). Can this also be the case of the one Izaña sample? The loss of the OH peak could also be explained by gypsum converting to anhydrite. If this is the case in your study then please change the conclusion. If not, then the discussion should cover why this volatile organic coating only applies to one sample. Isn't this expected from the other airborne samples too, at least the other Izaña samples?

We have added the XRD results of the unheated and heated Tenerife2014_1 sample (Figure 7) and refer to it on pg.13, ln. 7: *XRD analysis of the unheated and heated Tenerife2014_1 sample show the conversion of gypsum to anhydrite (Fig. 7).*

Furthermore, we have investigated the suggestion by the reviewer that needles may form when gypsum is transformed to anhydrite. We have compared SEM images of the Tenerife2014_1 sample before and after heating (now shown in Fig. 8 and in the supplementary). However, apart from one image (which we show in Fig. 8) we don't find any needles in the heated or unheated samples. Therefore, we don't think that the shape factor plays a role in our case. However, we now refer to the Grawe et al. 2018 study for the immersion mode ice nucleation activity of anhydrite. We use these data to discuss that anhydrite is more ice nucleation active than gypsum, which might be an explanation of the observed increase in $n_s$, which we find for the Tenerife2014_1 sample. However, because gypsum accounts only for 1 wt% and also needle formation is so limited, it seems unlikely to be the only factor. Therefore, we keep our other hypothesis of aliphatic compounds which is supported by the IR-ATR spectra and TGA measurements.

We have added and rephrased the following paragraph on pg. 13, ln.7:

*XRD analysis of the unheated and heated Tenerife2014_1 sample show the conversion of gypsum to anhydrite (Fig. 7). Gypsum has a low ice nucleation ability, similar to the clay minerals kaolinite and illite (Zimmermann et al., 2008). Grawe et al. (2018) found anhydrite to have a higher ice nucleation activity than quartz in the immersion mode at temperatures below 243 K when dry generated but a much lower $n_s$ when particles were generated from an aqueous solution. Anhydrite transforms back to gypsum when exposed to a relative humidity higher than 97% at room temperature (Bracconi et al., 2010). However, this process occurs on the order of hours to days, in line with the observed differences in $n_s$ between wet and dry generated particles in Grawe et al. (2018). Potentially, the transformation to anhydrite during heating explains the higher $n_s$ of the heated Tenerife2014_1 sample compared to the unheated one at subsaturated conditions. In this case, a partial conversion of anhydrite back to gypsum during RH conditions above water saturation might explain the unchanged $n_s$ of the unheated and heated Tenerife2014_1 sample above water saturation. It should be kept in mind that the bulk mineralogy as determined by XRD is not necessarily representative for the particle surface where ice nucleation takes place. As needle formation has been observed in the transformation of gypsum to anhydrite (Azimi and Papangelakis, 2011; Grawe et al., 2016), we use the occurrence of needles in our sample as an indication that the gypsum - anhydrite transformation took place on the surface of particles and thus might be responsible for the change in ice nucleation behavior of the sample. In SEM images of the unheated Tenerife2014_1 sample (Fig. 8a) hardly any needles are visible. A small number of needles is observed at the center of the image of the heated sample (Fig. 8b) while no needles are found in other SEM images of*

*the heated sample (see supplementary material). The apparently limited needle formation and the fact that only about 1 wt% gypsum is contained in the sample, suggests that gypsum transformation under heat treatment should only have small effect on the ice nucleation behavior of the Tenerife2014_1 sample. However, given that at maximum only about 10 % of the particles act as INPs in case of the Tenerife2014_1 sample, the gypsum-anhydrite transformation might be non-negligible. Additionally, we suggest that the increase in $n_s$ found under subsaturated $RH_w$ conditions for the Tenerife2014_1 sample is caused by the volatilization of aliphatic compounds containing matter, as indicated by the IR-ATR and TGA measurements, which inhibited the active sites of the mineral dust itself.*

Furthermore, we have amended our Conclusions (pg. 14, ln. 12):

*Three potential explanations are found, two of them related to changes in the mineralogy: While it cannot be excluded that the increase in $n_s$ was caused by a change in lattice spacing due to interlayer water release, it seems more likely that gypsum transforming to anhydrite made the sample more ice nucleation active. The Tenerife2014_1 sample is the only gypsum-containing sample that was investigated after heating, thus it remains an open question if and how much anhydrite contributed to the increase in $n_s$. Another reason for the increase could be that the ice nucleation active sites of the unheated sample were blocked by volatile organic material.*

**Technical comments:**

Title: Change '9' to 'nine' to be consistent with the partner paper.

Done

'Ice nucleating particles' without hyphen, like in partner paper and in Vali et al. (2015).

Done

Figure 1: The black color of the CALIMA2014 sample symbols covers all the other samples. Please change this to make it easier for the readers to see all results. Please remind the reader that CALIMA is the same sample location as Izaña in the figure legend.

We have re-sorted the layers of Fig. 1 such that the CALIMA results are in the background. Furthermore, we added in the caption of Fig. 1: *"which took place at the Izaña observatory"*

Figure 2: Please add to the legend text explaining the astrix (as in table 2). Why is the Etosha sample not present in this figure?

We have added in the caption of Fig. 2 the following sentence:

*The asterix in the legend indicates that the correlation is significant at the 0.05 level.*

And on Pg. 9, ln. 2: *The Etosha sample was excluded from the correlation as it does not contain any significant amount of these minerals.*

Figure 3: Please remind the reader which samples are airborne, milled and sieved in the figure legend. Why is the Atacama sample called milled and not the Australia and Morocco sample? Why does the heated sample from Morocco only have an upper limit?

We added in the caption of Fig. 3: *Square symbols indicate surface-collected samples, stars milled samples, and circles indicate airborne samples.*

The Atacama sample is called milled and not the Australia and Morocco samples because there is also an Atacama sieved sample while there aren't any other Australia or Morocco samples.

The lower error bars of the Morocco sample were switched off unnoticed. The same figure with the error bars included is now plotted.

Furthermore, we also changed the markers in Fig. 2, the same way as they are in Fig. 3.

Figure 5: In the text, the samples are discussed in the following order (ref. P10, L5-14) Australia, Etosha and Izaña 2014_2. The two first show no change between unheated and heated, and in the last sample a change is discussed. It would be more logical for the reader if the locations appear in the same order as the text.

We have reordered Fig. 5 such that a) Australia, b) Etosha and c) Tenerife2014_2.

Figure 6: The figure contains a lot of information, but the grey maps (1) add no important information as the images are taken at relatively low magnification so the particles can't really be seen. Also, please remove: "see text for details".

We have deleted the grey maps in Fig. 6 and removed "see text for details" in the caption.

Table 2: Please define the K-feldspar and feldspars groups in the figure legend. ˇ

Please explain the readers why there are different numbers of samples.

We have added to the caption of Table 2 the following sentence:

*K-feldspar comprises microcline and orthoclase, while feldspar refers to the sum of microcline, orthoclase and plagioclase. The number of samples included in each correlation varies because the $n_s$ of the Mojave, Peloponnese and Tenerife2014_2 samples was below the detection limit at 242 K, and the size distribution measurements of the Tenerife2014_1 sample were corrupted for the RH-scan at 238 K.*

**Introduction:**

Page 2, line 21: Please add a comma after the South Pole.

Done

Page 2, line 24: The term "potassium feldspars" is used sometimes, and "K-feldspars" other times. Please be consistent.

We have changed potassium feldspar to *K-feldspar*.

Page 3, line 31: Please specify the minerals in the K-feldspar fraction.

We have added: *, i.e. the fraction of microcline plus orthoclase,* behind K-feldspar fraction.

Page 3, line 33: Please specify the minerals in the feldspar fraction.

We have added: *(microcline, orthoclase and plagioclase)* behind feldspar fraction.

**Methods:**

Page 4, line 23: Please add country (Crete and Peloponnese, Greece) and then use the same structure as before, e.g. "… (Crete and Peloponnese, Greece), and the 10th of May 2010 (Aburdees, Egypt)."

Done

Page 5, line 16: "… mobility diameter between 12.2 – 615 nm…" Is the decimal place significant?

We agree that the decimal place is not of interest in this context and leave it out.

Page 6, line 25: Change 'tank' to 'sample container'

We decided to leave the word 'tank' here because this is how we describe the huge chamber where the aerosol is dispersed it, earlier in the manuscript and also in earlier publications. We feel that 'sample container' gives the impression of a much smaller volume.

**Results and discussion:**

Page 8: line 26 & 27: Please add p-value.

Done

Page 9, line 4: Please explain why these samples are selected.

We have changed

Pg. 9, ln. 4:  Figure 3 shows the $n$s at 240 K and 242 K of the native, i.e. unheated, and the heated Australia, Atacama milled, Izaña2014_2, Peloponnese, Etosha, and Morocco samples.

to:

Pg. 9, ln 23: *In this section, the role of heat labile material on the surface of dust particles is investigated. A representative subset of the samples was selected to reduce the number of experiments necessary. The Australia and Morocco sample were selected because of their exceptional high $n_s$, the Etosha sample, because its mineralogy did not explain the observed $n_s$, the Atacama milled sample because we expected a higher $n_s$ from its mineralogy, and finally the Tenerife2014_1 and Peloponnese samples as representatives for two airborne samples from different locations. Figure 3 shows the $n_s$ at 240 K and 242 K of the unheated and the heated samples.*

Page 10, line 6: Please remind the reader that the spectra are from bulk material. E.g. "… related to the dominant minerals (in bulk) in the samples".

We have changed this sentence to (Pg.10, Ln.31): *to the dominant minerals in the (bulk) samples*

Page 11, line 7 & 8: "The minerals contained in the Etosha sample… " Please remind the reader which minerals so they don't have to look it up in the partner paper.

We have added *(i.e. ankerite, calcite, dolomite, and muscovite)* behind 'Etosha sample' (now p. 12, ln.1)

P 11, line 9: Sentence too long.

We have split the sentence. Now Pg. 12, ln. 2: *In case of ankerite the ice nucleation ability is unknown. Based on its similarity with dolomite, a carbonate known not to be ice nucleation active, it is assumed that ankerite is also not active.*

Page 11, line 10: In this line you abruptly move from the discussion of minerals to organic material.

We have added on Pg.12, ln.3: *Thus, one of the suggested candidates with the strong Raman signal at 3180 cm$^{-1}$ is likely responsible for the ice nucleation activity of the Etosha sample.*

Page 11, line 33: Remove 'the' to During daytime,…

Done

**References:**

Azimi, G., and Papangelakis, V. G.: Mechanism and kinetics of gypsum–anhydrite transformation in aqueous electrolyte solutions, Hydrometallurgy, 108, 122-129, 10.1016/j.hydromet.2011.03.007, 2011.
Grawe, S., Augustin-Bauditz, S., Clemen, H. C., Ebert, M., Eriksen Hammer, S., Lubitz, J., Reicher, N., Rudich, Y., Schneider, J., Staacke, R., Stratmann, F., Welti, A., and Wex, H.: Coal fly ash: linking immersion freezing behavior and physicochemical particle properties, Atmos. Chem. Phys., 18, 13903-13923, 10.5194/acp-18-13903-2018, 2018.
Vali, G., DeMott, P. J., Möhler, O., and Whale, T. F.: Technical Note: A proposal for ice nucleation terminology, Atmos. Chem. Phys., 15, 10263-10270, 10.5194/acp-15-10263-2015, 2015.

---

## Author Comment (AC2) · 22 Dec 2018

We thank Reviewer 2 for their very constructive comments. We reproduce reviewer comments in blue in the following. Amended versions of the paper are given in *italics* for new sections and red text for the original text.

Boose et al. reports measurements of the effectiveness with which a range of natural dust samples from around the world to nucleate ice from supersaturated water vapour. Various analytical techniques were used to determine the composition of the samples and to attempt to establish what components are responsible for observed ice nucleation, leading to the conclusion that quartz and feldspar content is responsible for the ice nucleation observed in most samples. In one sample organic substances were found to be largely responsible while in another removal of organics actually enhanced ice nucleation effectiveness. The conclusions of the paper are interesting and relevant to the scope of ACP, highlighting the substantial complexity in ice nucleation by natural samples. The paper is mostly well written, sensibly structured, and is entirely suitable for publication in ACP, after a few minor corrections

**Minor comments**

I have a few issues with the identification of feldspar phases via powder XRD in this work. Alkali feldspar structure is a complex topic and it is not clear to me that powder diffraction is adequate to certain about which polymorph is present in dust samples of the type characterised here. Indeed in part 1 of this study the authors state 'Where microcline and orthoclase are present in the same sample, their individual fraction could not be distinguished reliably' in the caption to the relevant table. It is not clear to me what has changed in the present study.

The analysis of XRD used in the present study is the same as used in part 1 (Boose et al. 2016c). Nothing has changed and the same uncertainty applies here as well.

The difference between orthoclase and microcline is essentially the degree of order of the aluminosilicate framework. There is not necessarily a hard line between orthoclase and microcline. Parsons et al (American Mineralogist (2015) 100 (5-6), 1277-1303) and references therein discuss some of the relevant issues. Additionally the differences between even very pure samples of the two minerals in powder XRD are subtle. It must be more difficult to be certain of phase when mixed dusts with very low feldspar contents are under investigation. In summary, if the statements in the paper regarding the relative amounts of orthoclase and microcline are to be kept a much better description of the powder XRD procedure, and justification the differentiation between orthoclase and microcline is needed. I do not think this is particularly key to the paper, because, as the authors say, there is not any difference between the feldspars as characterised anyway. I would suggest treating the topic as it was treated in part 1 of this study.
Relatedly, Pg 8 line 12 states that microcline is the more active K-feldspar polymorph without citation. I think this statement should be changed, or its origin cited and discussed. The superior ice nucleating ability of microcline seems to be assumed throughout the paper. I would note that Whale et al. (Phys. Chem. Chem. Phys., (2017), 19, 31186) which used pure feldspars '...found no correlation between ice-nucleating efficiency and the crystal structures or the chemical compositions...' of ice nucleation active feldspars, albeit in rather different conditions.

We thank the reviewer for these valuable comments. We have decided to take out the sentences which suggest microcline being more active than orthoclase. We had based this statement on earlier studies by Augustin-Bauditz et al. (2014), GRL (doi: 10.1002/2014GL06131) and Harrison et al. (2016), ACP (doi: 10.5194/acp-16-10927-2016) but admit that the study by Whale et al. (2017) proves this to not hold in general, which we had overlooked.

We have made the following changes to the manuscript:

Pg.2, ln.27: Amongst the K-feldspars, microcline has a remarkable ice nucleation ability at temperatures up to 271 K (Harrison et al., 2016). However, Kaufmann et al. (2016) found microcline only in one out of eight dust samples collected in potential atmospheric dust source regions in South America, Asia and Africa and microcline was only present in one out of four airborne Saharan dust samples studied by Boose et al. (2016c).

Was changed to:

Pg. 2, ln.27: *While Kaufmann et al. (2016) found K-feldspar only in one out of eight dust samples collected in potential atmospheric dust source regions in South America, Asia and Africa, we observed K-feldspar to be present in all but one sample from a collection of 21 samples from deserts around the world (Boose et al., 2016c).*

Pg.8, ln.9:  Microcline was found in four out of five investigated Izaña samples (4-5 wt%) and orthoclase (4-5 wt%) is present in one Izaña and the other three Saharan samples. Comparing the bulk mineralogy to the surface-dependent ice nucleation ability has the caveat to introduce uncertainty. Within this uncertainty there is no detectable effect from the presence of the more ice nucleation active microcline K-feldspar over orthoclase for the airborne samples at the studied temperatures.

was changed to:

Pg. 8, ln.21: *Differentiating between microcline and orthoclase and comparing the bulk mineralogy to the surface-dependent ice nucleation ability, introduces uncertainty. Within this uncertainty, there is no detectable effect from the presence of microcline versus orthoclase in the airborne samples at the studied temperatures. This is in line with the findings by Whale et al. (2017) that there is no correlation between ice nucleating ability and the level of ordering in the aluminosilicate framework, which determines if orthoclase (less ordered) or microcline (more ordered) is present.*

And we deleted the following from the initial manuscript:

Pg. 12, ln. 30: Furthermore, we observed that the ice nucleation activity of airborne Saharan dust at these temperatures is not higher when microcline is present in the samples compared to when orthoclase is present at typical amounts of 4-5%. If one assumes that both feldspars are equally active at these low temperatures, this result is to be expected.

Naming of samples is not entirely consistent through the manuscript I think, this should be checked.

We changed "native" to "unheated" on originally Pg. 9, ln 4 and Pg. 12, ln12

Furthermore, we have changed the naming of the Izaña201x_x samples into Tenerife201x_x and have numbered the 2014 samples from 1 to 3 to be consistent with part 1 of this study:

Izaña2013_2 ➔ Tenerife2013
Izaña2014_2 ➔ Tenerife2014_1
Izaña2014_3 ➔ Tenerife2014_2
Izaña2014_5 ➔ Tenerife2014_3

**Specific comments**
Abstract line 7 to 9- sentence starting 'in this study,....' does not read well.

We have changed the sentence to:
Pg.1, ln.7: *In this study, the influence of semi-volatile organic compounds and the presence of crystal water on the ice nucleation behavior of desert aerosol is investigated.*

Abstract line 9- between 238 and 242K
Changed

Abstract line 15 reads poorly, missing comma after 'diminished'?
Comma was added after 'diminished'

Pg 2 Line 2- delete 'already'
Done

Pg 2 Line 25- ACP version of Paramonov et al. is available
Reference was changed to ACP version

Pg 3 line 32- clumsy wording, probably delete 'showing'
We replaced 'showing' with 'found'. The sentence now reads:
Pg.3, ln.31: *At T ≤ 245 K the best correlation of the ice nucleation activity was found for the bulk quartz plus feldspar content in the dust samples while the fraction of clays was negatively correlated with the ice nucleation activity.*

Pg 3 line 10- typo 'selected'
Corrected

Pb 3 line 33- clumsy sentence structure
Pg 3 line 35- 'over' not the right work I think.
The respective sentence

Quartz alone has been found to show various immersion mode ice nucleation activities in laboratory studies, ranging from being active at temperatures comparable to microcline (Zolles et al., 2015), over being active below feldspar temperatures but above those of clay (Atkinson et al., 2013), to showing ice nucleation activity at temperatures comparable or lower than those of clays (Kaufmann et al., 2016).
has been replaced by:

Pg.3, ln.34: *Quartz alone has been found to show various immersion mode ice nucleation activities in laboratory studies. Zolles et al. (2015) found quartz being active at temperatures comparable to microcline, while Atkinson et al. (2013) measured ice nucleation activity below feldspar temperatures but above those of clay. Kaufmann et al. (2016), on the other hand, only observed ice nucleation activity at temperatures comparable to or lower than those of clays.*

Pg 3 line 33 onwards- I note that there is interesting work on the topic of ice nucleation by quartz under review in ACPD at current (https://www.atmos-chem-physdiscuss.net/acp-2018-1020/), which may shed some light on the complexities of ice nucleation by various silicas.
We have added the following sentence on p. 4, ln. 2:
*These differences in ice nucleation ability can be related to the history of the quartz samples and different ways of pre-processing them (Zolles et al., 2015).Milling quartz samples leads to a break-up of Si-O-Si bridges on the surface, leading to the formation of Si-OH and Si-O-OH in the presence of water vapor, which increases the ice nucleation activity of the quartz particles (Kumar et al., 2018b).*

Further, we discuss our results now in the light of the study by Kumar et al. (2018b) in the Conclusions.
Pg. 12, ln.27: It remains unknown why laboratory studies show various ice nucleation activities of pure quartz particles (Atkinson et al., 2013; Zolles et al., 2015; Kaufmann et al., 2016), relating to the question of how ice nucleation occurs on dust in general and on quartz in particular.
Was changed to:

Pg. 14, ln. 2: *According to a recent study by Kumar et al. (2018b), the variation in quartz ice nucleation ability found in laboratory studies (Atkinson et al., 2013; Zolles et al., 2015; Kaufmann et al., 2016) and the superior ice nucleation ability of the quartz-rich samples from Australia and Morocco in this study and its partner paper, may be explained by the pre-processing of the samples. Milling of quartz samples, as done in our study, increases the ice nucleation ability of quartz by creating Si-O• and Si• radical sites which can then react with water vapor (Kumar et al., 2018b). However, milling may not be the only reason for formation of the silanol (Si-OH) groups on the surface of quartz, because exposure to water molecules in ambient humidity could also result in passively converting surface siloxane groups (Si-O-Si) to silanol groups (Boehm, 1966; Wang et al., 2018). As such quartz samples may still exhibit high ice nucleation activity in the absence of milling due to the particles chemical history. Thus, it remains an open question, how much quartz contributes to the ice nucleation ability of (unmilled) atmospheric dust.*

Pg 4 line 7- while RH is increased
Changed

Pg 8 line 15- Some justification for using the conditions stated might aid the reader.
The original sentence:
This comparison showed that the immersion mode ice nucleation activity at T ≤ 245 K correlates best

with the quartz and quartz+feldspar fractions of the dust samples, while at T = 253 K it correlates best with the K-feldspar fraction alone (Boose et al., 2016c).

was changed to:

Pg. 8, ln.27: *This comparison showed that the immersion mode ice nucleation activity correlates best with the K-feldspar fraction alone at T = 253 K, a temperature where only feldspar minerals are found to have significant ice nucleation activity (Atkinson et al. 2013). At T ≤ 245 K, the ice nucleation activity correlates best with the quartz and quartz + feldspar fractions of the dust samples (Boose et al., 2016c). At these lower temperatures, quartz and to a lower degree also clay minerals were found to nucleate ice efficiently (Atkinson et al. 2013).*

Pg 11 line 15- I'm not sure what is meant by the sentence 'However, it was not....'. I suggest clarifying this.

We have deleted the sentence. We just wanted to stress that ammonium sulfate alone is not responsible for ice nucleation of dust. However, this is likely obvious for the reader and the sentence rather confusing.

Pg 9 line 15- I am not sure what is meant by 'hardly' in this context? This sentence could be clearer.

We believe the reviewer is referring to pg. 9 ln. 17 of the initial manuscript. We have now changed this sentence from

However, the complexity of the dust samples in this study, which consist of several minerals and likely also various other components, cause the spikes to widen and hardly discrete steps to be observed.

to now pg. 10 ln. 5:

*However, the complexity of the dust samples in this study, which consist of several minerals and likely also various other components, cause the spikes to widen thus reducing the possibility to observe discrete steps.*

Pg 11 line 12- remove comma after 'both'

Done

Pg 12 line 6- brackets around '2006' after 'Laird'

Done

Pg 12 line 10- sentence starting 'Thus, according to our observations...' is clumsy

The respective sentence

Thus, according to our observations, the collapse of the smectite lattice should, if at all, only influence the Izaña2014_2 sample and then be related to an increase in ice nucleation ability.

was changed to:

Pg.13, ln.4: *Thus, the collapse of the smectite lattice should only influence the Tenerife2014_1 sample. In case it has an influence, this would be related to an increase in ice nucleation ability.*

Figure 2- It is not obvious what the * in 0.63* refers to without referring to final table.

We added a sentence in the caption of Fig. 2 for clarification:

*The asterix in the legend indicates that the correlation is significant at the 0.05 level.*